# Divergent acyl carrier protein decouples mitochondrial Fe-S cluster biogenesis from fatty acid synthesis in malaria parasites

Seyi Falekun[1], Jaime Sepulveda[1], Yasaman Jami-Alahmadi[2], Hahnbeom Park[3], James A Wohlschlegel[2], Paul A Sigala[1]*

[1]Department of Biochemistry, University of Utah School of Medicine, Salt Lake City, United States; [2]Department of Biological Chemistry, University of California, Los Angeles, Los Angeles, United States; [3]Department of Biochemistry, University of Washington, Seattle, United States

*For correspondence: p.sigala@biochem.utah.edu

Competing interest: The authors declare that no competing interests exist.

**Abstract** Most eukaryotic cells retain a mitochondrial fatty acid synthesis (FASII) pathway whose acyl carrier protein (mACP) and 4-phosphopantetheine (Ppant) prosthetic group provide a soluble scaffold for acyl chain synthesis and biochemically couple FASII activity to mitochondrial electron transport chain (ETC) assembly and Fe-S cluster biogenesis. In contrast, the mitochondrion of *Plasmodium falciparum* malaria parasites lacks FASII enzymes yet curiously retains a divergent mACP lacking a Ppant group. We report that ligand-dependent knockdown of mACP is lethal to parasites, indicating an essential FASII-independent function. Decyl-ubiquinone rescues parasites temporarily from death, suggesting a dominant dysfunction of the mitochondrial ETC. Biochemical studies reveal that *Plasmodium* mACP binds and stabilizes the Isd11-Nfs1 complex required for Fe-S cluster biosynthesis, despite lacking the Ppant group required for this association in other eukaryotes, and knockdown of parasite mACP causes loss of Nfs1 and the Rieske Fe-S protein in ETC complex III. This work reveals that *Plasmodium* parasites have evolved to decouple mitochondrial Fe-S cluster biogenesis from FASII activity, and this adaptation is a shared metabolic feature of other apicomplexan pathogens, including *Toxoplasma* and *Babesia*. This discovery unveils an evolutionary driving force to retain interaction of mitochondrial Fe-S cluster biogenesis with ACP independent of its eponymous function in FASII.

## Introduction

Malaria is an ancient scourge of humanity and remains a pressing global health challenge, especially in tropical Africa where hundreds of thousands of people die from malaria each year. *Plasmodium falciparum* malaria parasites are single-celled eukaryotes that evolved under unique selective pressures with unusual metabolic adaptations compared to human cells and well-studied model organisms such as yeast. Understanding the unique biochemical pathways that specialize parasites for growth within human red blood cells will shed light on their evolutionary divergence from other eukaryotes and unveil new parasite-specific targets for development of novel antimalarial therapies.

*P. falciparum* retains an essential mitochondrion required for biosynthesis of pyrimidines, acetyl-CoA, and Fe-S clusters (*Painter et al., 2007*; *Jhun et al., 2018*; *van Dooren et al., 2006*; *Oppenheim et al., 2014*; *Gisselberg et al., 2013*). Although the parasite mitochondrion also contains enzymes involved in the citric acid cycle and biosynthesis of ATP and heme, these pathways are dispensable for blood-stage parasites, which can scavenge host heme and obtain sufficient ATP from cytoplasmic

glycolysis (*Ke et al., 2014*; *Nagaraj et al., 2013*; *Ke et al., 2015*; *Sturm et al., 2015*; *MacRae et al., 2013*). In addition to these pathways, most eukaryotes, including mammals and yeast, also contain a mitochondrial type II fatty acid biosynthesis (FASII) pathway that generates the octanoate precursor of the lipoic acid cofactor used by several mitochondrial dehydrogenases (*Hiltunen et al., 2009*). In contrast to human and yeast cells, FASII enzymes in *P. falciparum* have been lost by the mitochondrion and are retained instead by the apicoplast organelle (*Figure 1—figure supplement 1*; *Shears et al., 2015*). Although critical lipoate-dependent enzymes are present in the parasite mitochondrion (*Jhun et al., 2018*), prior work has shown that these enzymes utilize scavenged lipoate obtained from the red blood cell rather than de novo synthesis (*Allary et al., 2007*).

The acyl carrier protein (ACP) is a key component of FASII and contains a strictly conserved Ser residue modified by a 4-phosphopantetheine (Ppant) group whose terminal thiol tethers the nascent acyl chain during fatty acid initiation, modification, and elongation (*Chan and Vogel, 2010*). Consistent with FASII targeting to the apicoplast in *P. falciparum*, this organelle features a well-studied ACP homolog (PF3D7_0208500, aACP) that retains canonical ACP features, including the conserved Ser modified by a 4-Ppant group required for FASII function (*Shears et al., 2015*; *Gallagher and Prigge, 2010*). Although apicoplast FASII activity is essential for *P. falciparum* growth within mosquitoes and the human liver, this pathway is dispensable for blood-stage parasites, which can scavenge fatty acids from the host (*Shears et al., 2015*; *van Schaijk et al., 2014*; *Yu et al., 2008*).

Despite loss of mitochondrial FASII enzymes, *P. falciparum* has curiously retained a second ACP homolog (PF3D7_1208300) that is annotated as a mitochondrial ACP (mACP) but has not been directly studied. We noted that recent genome-wide knock-out studies in *Plasmodium berghei* and *P. falciparum* both reported that this gene was refractory to disruption, suggesting an essential, FASII-independent function (*Zhang et al., 2018*; *Gomes et al., 2015*). We became interested to unravel what this essential function might be. Recent studies in yeast and human cells have identified an expanding network of mitochondrial ACP interactions beyond FASII that includes roles in respiratory chain assembly, Fe-S cluster biogenesis, and mitochondrial ribosomal translation (*Van Vranken et al., 2016*; *Van Vranken et al., 2018*; *Nowinski et al., 2020*; *Brown et al., 2017*; *Majmudar et al., 2019*). However, in all cases these critical interactions in yeast and/or humans are linked to mitochondrial FASII function and the Ppant group of ACP (*Brown et al., 2017*; *Cory et al., 2017*; *Angerer et al., 2014*; *Boniecki et al., 2017*).

We localized mACP to the *P. falciparum* mitochondrion and set out to understand its FASII-independent function in this organelle. Using conditional knockdown and immunoprecipitation (IP) studies, we discovered that mACP is essential for parasite viability and plays a critical role in binding and stabilizing the Isd11-Nfs1 cysteine desulfurase complex required for mitochondrial Fe-S cluster biogenesis. Unlike mACP interactions in yeast and humans, *P. falciparum* mACP binds to Isd11 via a divergent molecular interface that does not involve a 4-Ppant group. This work unveils a new molecular paradigm for essential mACP function without a Ppant group, underscores the critical and conserved role of mACP in mitochondrial Fe-S cluster biogenesis, and highlights a *Plasmodium*-specific adaptation suitable for exploration as a metabolic vulnerability for parasite-directed antimalarial therapy.

## Results

### Mitochondrial ACP is essential for *P. falciparum* despite loss of FASII in this organelle

The *P. falciparum* genome encodes two ACP homologs that include the well-studied protein targeted to the apicoplast (aACP, PF3D7_0208500) and a second homolog annotated as a mitochondrial ACP (mACP, PF3D7_1208300). Unlike the apicoplast ACP, which retains the conserved Ser for 4-Ppant attachment, mACP has curiously replaced this Ser with a Phe residue that cannot be modified by a 4-Ppant group (*Figure 1A* and *Figure 1—figure supplement 2*). This Ser-to-Phe substitution in mACP is consistent with the loss of mitochondrial FASII and the ACP-modifying phosphopantetheine transferase enzyme (*Figure 1—figure supplement 1*) and suggested a non-canonical function for mACP.

To test if mACP localized to the parasite mitochondrion, we created a Dd2 *P. falciparum* line that episomally expressed mACP fused to a C-terminal dual hemagglutinin (HA) tag. Immunofluorescence analysis of this mACP-HA$_2$ line revealed strong colocalization between mACP-HA$_2$ and the mitochondrial marker, HSP60 (*Figure 1B*, *Figure 1—figure supplement 3*). On the basis of this colocalization,

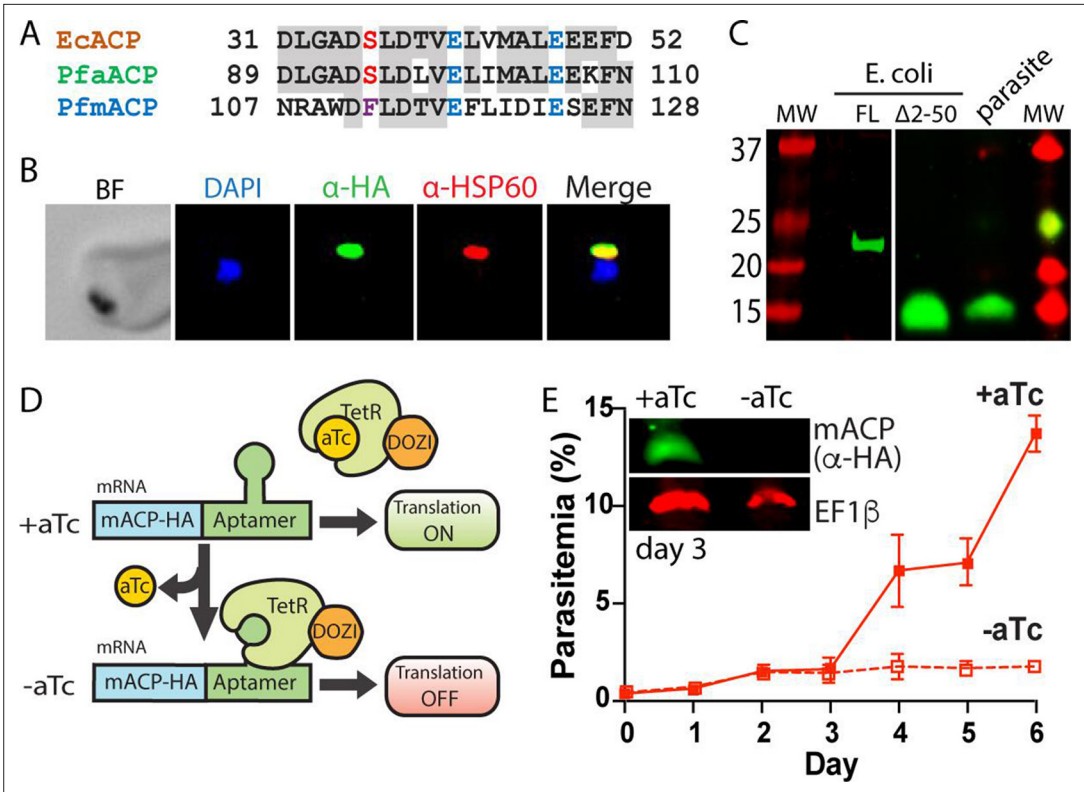

**Figure 1.** Mitochondrial ACP is essential for blood-stage *P. falciparum*. (**A**) Partial sequence alignment of ACP homologs from *E. coli* and the *P. falciparum* apicoplast (aACP) and mitochondrion (mACP). The conserved Ser residue that is modified by a 4-Ppant group is in red, the divergent Phe residue of mACP is in purple, and conserved Glu residues are in blue. (**B**) Immunofluorescence microscopy of fixed Dd2 parasites episomally expressing mACP-HA$_2$ and stained with DAPI (nucleus, blue), anti-HA (green), and anti-HSP60 (mitochondrion, red) antibodies (BF = bright field). (**C**) Anti-HA western blot (WB) analysis of cellular lysates of *E. coli* expressing full-length (FL) or truncated (Δ2–50) mACP-HA$_2$ and Dd2 *P. falciparum* parasites episomally expressing full-length mACP-HA$_2$. The left and right images are from different WB experiments (10% gels) that were aligned by molecular weight (MW) markers. (**D**) Schematic depiction of protein translational regulation using the aptamer/TetR-DOZI system. Normal protein expression occurs in the presence but not absence of anhydrotetracycline (aTc). (**E**) Continuous growth assay of synchronous mACP-HA/FLAG-aptamer/TetR-DOZI parasites in the presence or absence of aTc. Data points and error bars are the average and standard deviation from two biological replicates. Inset is an anti-HA and anti-EF1$\beta$ (37 kDa) WB of parasite lysates harvested on day 3 of the continuous growth assay. WBs were repeated 2–3 times.

The online version of this article includes the following source data and figure supplement(s) for figure 1:

**Figure supplement 1.** *P. falciparum* homologs of *E. coli* FASII proteins.

**Figure supplement 1—source data 1.** Source file for *Figure 1—figure supplement 1*.

**Figure supplement 2.** Sequence and mass spectrometry (MS) analyses of ACP.

**Figure supplement 3.** Additional immunofluorescence microscopy images mACP-HA$_2$.

**Figure supplement 4.** Schematic depiction of mACP gene editing and verifying genomic integration by PCR.

**Figure supplement 5.** Giemsa-stained blood smears of mACP-aptamer/TetR-DOZI parasites cultured for 120 hr (5 days) ± anhydrotetracycline (aTc).

we conclude that mACP is indeed targeted to the parasite mitochondrion, as predicted. Full-length mACP-HA$_2$ has a predicted molecular mass ~21 kDa, but SDS-polyacrylamide gel electrophoresis (SDS-PAGE) and western blot (WB) analysis of immunoprecipitated mACP-HA$_2$ indicated that this protein migrated with an apparent molecular mass ~15 kDa (*Figure 1C*). This lower size suggested that mACP was likely to be post-translationally processed upon mitochondrial import, which would be consistent with known processing of mitochondrial ACP in yeast and humans (*Vögtle et al., 2009*; *Vaca Jacome et al., 2015*). We extended our WB analysis to include mACP-HA$_2$ that was expressed

heterologously in *Escherichia coli*, where proteolytic processing is not expected. The recombinant, full-length mACP-HA$_2$ expressed in bacteria migrated with an apparent molecular mass close to 21 kDa, as expected. Based on sequence homology, the known ACP-processing sites in yeast and humans (*Vögtle et al., 2009*; *Vaca Jacome et al., 2015*), and mass spectrometry (MS) analysis of parasite-expressed mACP (*Figure 1—figure supplement 2*), we identified Leu-51 as the possible N-terminus of mature mACP in *P. falciparum*. We therefore cloned and bacterially expressed a truncated (Δ2–50) mACP-HA$_2$ beginning with Leu-51 (after a start Met). This truncated mACP protein co-migrated by SDS-PAGE with mACP-HA$_2$ expressed in parasites (*Figure 1C*), suggesting that Leu-51 is at or near the N-terminus of mature mACP in parasites. We conclude that mACP is proteolytically processed upon import into the *P. falciparum* mitochondrion.

To directly test if mACP is essential for blood-stage parasite growth and viability, we used CRISPR/Cas9 to tag the mACP gene in Dd2 parasites to encode a C-terminal HA-FLAG epitope tag and the aptamer/TetR-DOZI system that enables anhydrotetracycline (aTc)-dependent control of protein expression (*Figure 1D*; *Ganesan et al., 2016*). Correct integration into the mACP locus was confirmed by PCR analysis of polyclonal and clonal parasite lines (*Figure 1—figure supplement 4*). To evaluate mACP knockdown and its impact on parasite growth, we synchronized parasites to the ring stage, split these parasites into two equal populations ± aTc, and monitored parasite growth over multiple 48 hr growth cycles. Parasites grew indistinguishably ± aTc for the first 3 days. However, parasites grown without aTc displayed a major growth defect on day 4 in the third intraerythrocytic lifecycle (*Figure 1E*), similar to prior studies with this knockdown system (*Ganesan et al., 2016*). Blood-smear analysis on day 5 indicated widespread parasite death (*Figure 1—figure supplement 5*). WB analysis of parasite samples harvested in the second cycle, after 3 days of growth -aTc, indicated robust knockdown of mACP expression relative to parasites grown +aTc (*Figure 1E*). We conclude that mACP is essential for *P. falciparum* parasites during blood-stage growth.

## Mitochondrial ACP binds the Isd11-Nfs1 complex required for Fe-S cluster biogenesis

Recent studies in yeast and human cells have revealed that mitochondrial ACP binds to a variety of small, three-helical adapter proteins that bear a conserved Leu-Tyr-Arg (LYR) sequence motif on their N-terminal helix (*Majmudar et al., 2019*; *Angerer, 2015*). The Leu mediates an intramolecular helical-helical contact while the Tyr and Arg side chains interact with conserved acidic residues on ACP (*Cory et al., 2017*; *Boniecki et al., 2017*; *Herrera et al., 2019*). These LYR-motif proteins mediate diverse mitochondrial processes that include respiratory chain assembly, Fe-S cluster biogenesis, and ribosomal translation (*Van Vranken et al., 2016*; *Van Vranken et al., 2018*; *Brown et al., 2017*; *Angerer et al., 2014*). Given the strong conservation of mitochondrial LYR proteins and ACP interactions across eukaryotes, we reasoned that the essential function of *P. falciparum* mACP might involve interaction with a conserved LYR protein.

Using the sequences of known LYR proteins from yeast and humans as bait (*Angerer, 2015*), we conducted a BLAST search of the *P. falciparum* genome to look for possible homologs. In contrast to other eukaryotes, *P. falciparum* lacked identifiable homologs of LYR proteins that mediate respiratory chain assembly (LYRM3/NDUFB9, LYRM6/NDUFA6, LYRM7/MZM1, LYRM8/SDHAF1, ACN9/SDHAF3, and FMC1/C7orf55) or interactions with mitochondrial ribosomes (L0R8F8). However, we identified a clear homolog for Isd11 (PF3D7_1311000, e-value 3e-6), which functions in Fe-S cluster biogenesis. Our BLAST analysis suggested that Isd11 is the only LYR-protein homolog retained by parasites (*Figure 2—figure supplement 1*).

*P. falciparum* Isd11 is 29% identical to human Isd11 (also known as LYRM4), including conservation of the LYR sequence motif near the N-terminus (*Figure 2A*). The parasite mACP retains the conserved Glu residues (5 and 11 amino acids C-terminal to the modified Ser in canonical ACPs) known to interact with the Tyr and Arg residues of the LYR motif on Isd11 at the mACP-Isd11 binding interface (*Figure 1A*; *Cory et al., 2017*; *Boniecki et al., 2017*). In yeast and humans, Isd11 directly binds and stabilizes Nfs1, a mitochondrial cysteine desulfurase required for biogenesis of Fe-S clusters (*Wiedemann et al., 2006*). Mitochondrial ACP in these organisms directly binds Isd11 to stabilize the mACP-Isd11-Nfs1 complex (*Figure 2B*), with loss of ACP and/or Isd11 resulting in Nfs1 instability and defective Fe-S cluster biogenesis (*Van Vranken et al., 2016*; *Lill and Freibert, 2020*; *Herrera et al., 2018*; *Cai et al., 2017*). *P. falciparum* retains a mitochondrial Nfs1 homolog (PF3D7_0727200) that

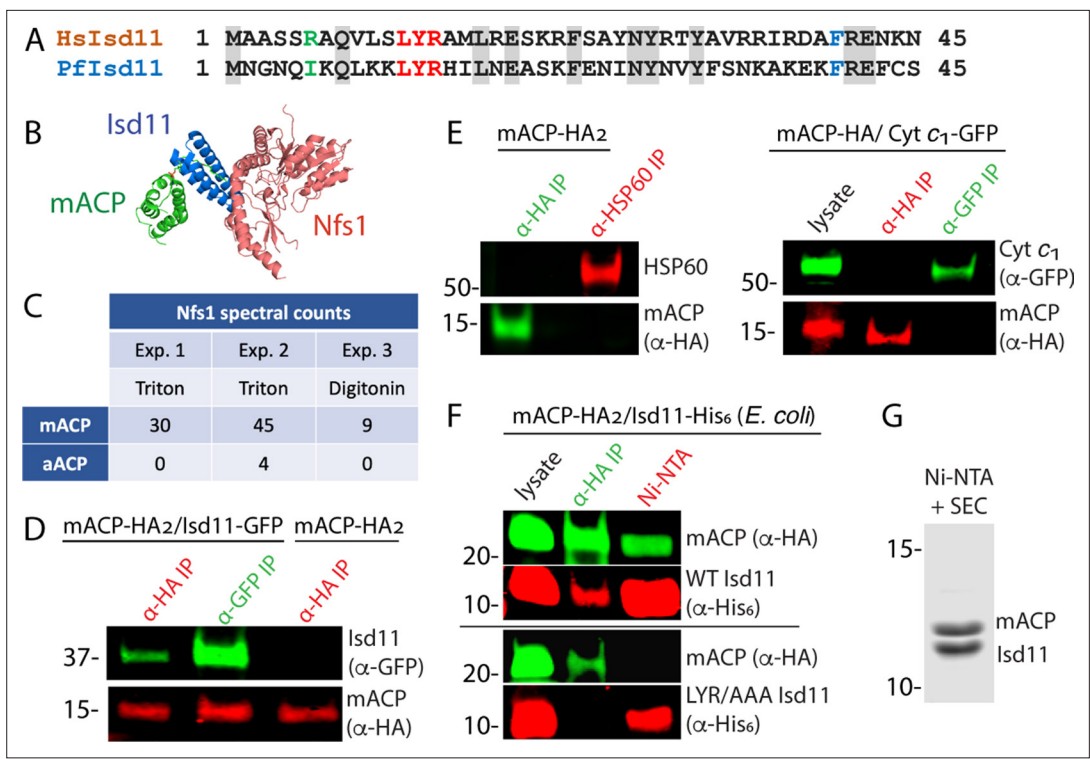

**Figure 2.** Mitochondrial ACP binds the Isd11-Nfs1 complex. (**A**) Partial sequence alignment of human and *P. falciparum* Isd11. The conserved LYR sequence motif is in red, the conserved Phe residue is in blue, and residue 6 is in green. (**B**) X-ray structural model of the mACP-Isd11-Nfs1 complex (PDB entry 5USR). For simplicity, only a single copy of each protein is shown, rather than the functional dimer (***Boniecki et al., 2017***). (**C**) Table of spectral counts for *P. falciparum* Nfs1 detected by tandem mass spectrometry for anti-HA immunoprecipitation (IP) studies of lysates from Dd2 parasites episomally expressing mACP-HA$_2$ or aACP-HA$_2$. Parasites were lysed in either Triton X-100 or digitonin. Similar spectral counts for mACP and aACP bait proteins were detected in each experiment (***Figure 2—figure supplement 5***). (**D**) Anti-HA or anti-GFP co-immunoprecipitation (co-IP)/western blot (WB) studies of Dd2 parasites episomally expressing mACP-HA$_2$ with or without Isd11-GFP. (**E**) Anti-HA, anti-HSP60, or anti-GFP co-IP/WB studies of Dd2 parasites episomally expressing mACP-HA$_2$ or endogenously expressing mACP-HA (knockdown line) and episomally expressing cyt $c_1$-GFP. (**F**) Anti-HA co-IP or nickel-nitrilotriacetic acid (Ni-NTA) pulldown and WB studies of *E. coli* bacteria recombinantly expressing full-length mACP-HA$_2$ and Isd11-His$_6$ (WT or LYR/AAA mutant) and probed with anti-HA and anti-His$_6$ antibodies. All lanes were loaded with equivalent sample volumes, are from the same blot, and were processed identically but were cropped from non-contiguous lanes. (**G**) Coomassie-stained SDS-PAGE gel of recombinant His$_6$-Isd11 and Δ2–50 mACP-HA$_2$ purified from *E. coli* by Ni-NTA pulldown of His$_6$-Isd11 and size-exclusion chromatography (SEC) after removal of affinity tags. Additional images for WB experiments are shown in ***Figure 2—figure supplement 3***. WBs are representative of 2–3 replicate experiments with independent samples.

The online version of this article includes the following source data and figure supplement(s) for figure 2:

**Figure supplement 1.** BLAST analysis of LYR-motif protein homologs found in *P. falciparum*.

**Figure supplement 1—source data 1.** Source file for ***Figure 2—figure supplement 1***.

**Figure supplement 2.** Specific enrichment of Nfs1 in anti-HA-tag IP of mACP-HA$_2$ relative to anti-HA-tag IP aACP-HA$_2$.

**Figure supplement 3.** Co-localization of Isd11-GFP and MitoTracker in Dd2 parasites expressing mACP-HA$_2$ and Isd11-GFP.

**Figure supplement 4.** Additional images for western blot (WB) analyses.

**Figure supplement 5.** Spectral counts for *P. falciparum* mACP or aACP detected by tandem mass spectrometry.

**Figure supplement 5—source data 1.** Source file for ***Figure 2—figure supplement 5***.

**Figure supplement 6.** Mass spectrometry (MS) analysis of recombinant parasite Isd11 and mACP expressed in bacteria.

**Figure supplement 7.** *P. falciparum* Isd11 does not stably bind to apicoplast ACP.

is predicted from genome-wide knock-out studies to be essential (*Gomes et al., 2015*; *Gisselberg et al., 2013*) but whose role in Fe-S cluster biogenesis has not been studied. Collectively, conservation of Isd11 and Nfs1 in *P. falciparum*, the conserved role for ACP in stabilizing the mACP-Isd11-Nfs1 complex in other eukaryotes, and retention of the LYR sequence motif in *P. falciparum* Isd11 strongly suggested that mACP was likely to interact with Isd11 to form the mACP-Isd11-Nfs1 complex in parasites.

To identify protein interaction partners of mACP, we used anti-HA IP to isolate mACP-HA$_2$ from parasites followed by tandem MS to identify parasite proteins that co-purified with mACP. In multiple independent pulldowns, we identified Nfs1 as strongly enriched in the mACP-HA$_2$ sample compared to a negative control sample (anti-HA IP of aACP-HA$_2$) (*Figure 2C*, *Figure 2—figure supplement 2*). We did not observe peptides corresponding to Isd11 in mACP-HA$_2$ IP/MS samples. However, its absence is not unexpected given the short Isd11 length (87 residues) that leads to comparatively few tryptic peptides.

To directly test for mACP interaction with Isd11, we transfected the mACP-HA$_2$/Dd2 line with an episome encoding expression of Isd11 with a C-terminal GFP tag that was previously used to localize Isd11 to the mitochondrion (*Gisselberg et al., 2013*). We confirmed that Isd11-GFP expressed in this line colocalized with MitoTracker (*Figure 2—figure supplement 3*). Using this Dd2 line expressing mACP-HA$_2$ and Isd11-GFP, we performed reciprocal IP/WB experiments that confirmed stable interaction of these two proteins in parasites (*Figure 2D*, *Figure 2—figure supplement 4*). This interaction appeared to be specific as mACP did not co-IP with the abundant mitochondrial chaperone, HSP60, or with GFP-tagged cytochrome $c_1$ (*Figure 2E*, *Figure 2—figure supplement 4*). Because these IP experiments could not distinguish whether mACP binding to Isd11 was direct or mediated by other parasite proteins, we turned to heterologous studies in *E. coli*. Reciprocal IP experiments with lysates from bacteria coexpressing *P. falciparum* mACP-HA$_2$ and Isd11-His$_6$ confirmed stable interaction of these proteins in the absence of other parasite-specific factors (*Figure 2F*). In experiments with Isd11 LYR mutants, however, this interaction was strongly reduced (YR/AA) or eliminated (LYR/AAA), despite similar expression of all recombinant proteins (*Figure 2F*, *Figure 2—figure supplement 4*). Clarified lysates from bacteria coexpressing truncated (Δ2–50) mACP-HA$_2$ and Isd11-His$_6$ were also fractionated by passage over Ni-NTA resin to selectively pull down His-tagged Isd11 and interactors followed by size-exclusion chromatography. Analysis by Coomassie-stained SDS-PAGE and tandem MS indicated robust isolation of a 1:1 complex containing Isd11 and mACP (*Figure 2G*, *Figure 2—figure supplement 6*). We conclude that mACP directly binds to Isd11 and that this interaction involves the LYR motif.

Yeast and human Isd11 retain a conserved Arg-6 residues upstream of the LYR motif (*Figure 2A*) that electrostatically stabilizes the negatively charged oxygen atoms of the Ppant group in the ACP-Isd11 complex (*Cory et al., 2017*; *Boniecki et al., 2017*; *Herrera et al., 2019*), and mutation of this Arg ablates binding of human Isd11 to ACP (*Majmudar et al., 2019*). Because parasite mACP lacks the negatively charged Ppant group and has replaced it with a hydrophobic Phe, retention of Arg-6 by *P. falciparum* Isd11 would be predicted to destabilize binding of these two proteins. The parasite Isd11, however, has replaced Arg-6 with an Ile residue (*Figure 2A*) expected to more favorably interact with the local hydrophobic surface of mACP created by Ser-to-Phe substitution. These sequence features suggest that parasite Isd11 has co-evolved with mACP to optimize formation of the mACP-Isd11 complex in *P. falciparum* in the absence of FASII and an acyl-Ppant group on mACP. Consistent with this model, recombinantly parasite Isd11 expressed in bacteria did not co-purify with the endogenous ~9 kDa *E. coli* ACP (*Figure 2G*) or with coexpressed apicoplast ACP (*Figure 2—figure supplement 7*), both of which retain the conserved Ser and Ppant modification. This specificity by *P. falciparum* Isd11 contrasts with that of human Isd11, which robustly binds bacterial ACP when expressed in *E. coli* (*Cory et al., 2017*; *Boniecki et al., 2017*; *Cai et al., 2017*).

## Structural modeling of the mACP-Isd11 interface

Association of Isd11 and mACP in yeast and human cells depends critically on the acyl-Ppant group of mACP (*Figure 3A*), with little or no association observed in these cells between Isd11 and the mACP Ser-to-Ala mutant that lacks a Ppant group (*Van Vranken et al., 2016*; *Van Vranken et al., 2018*; *Majmudar et al., 2019*). The mACP-Isd11 interface in these organisms also includes key electrostatic interactions between conserved Lys and Arg residues in Isd11 and Glu and Asp residues in

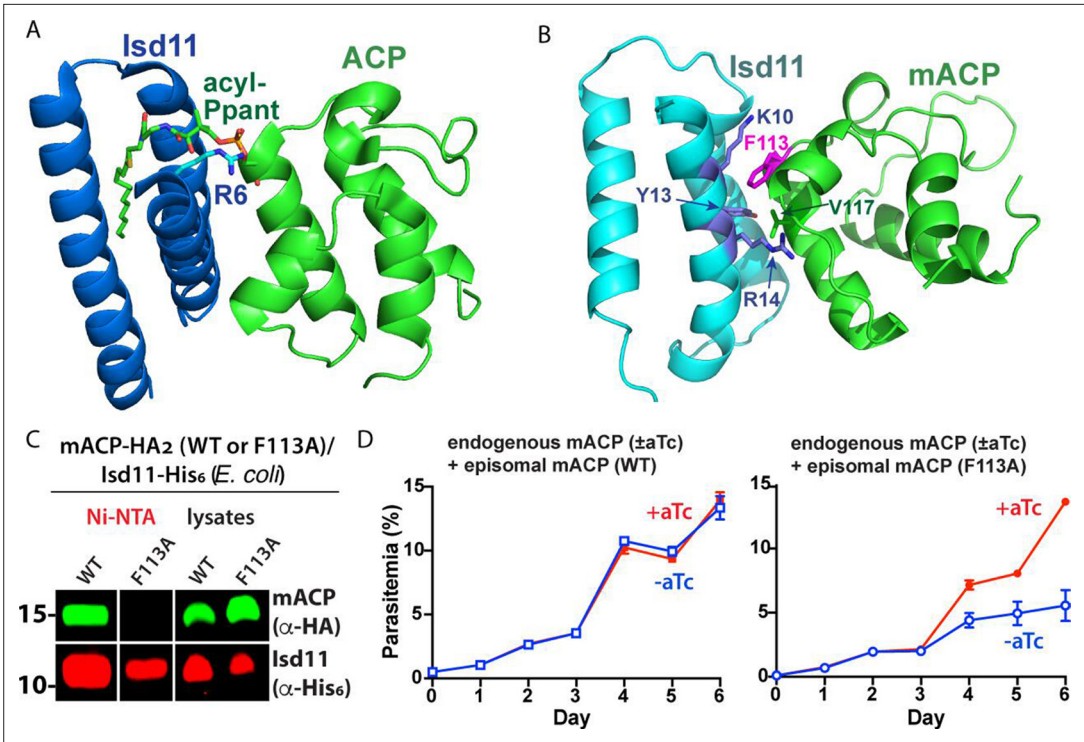

**Figure 3.** Structural modeling and functional tests of the divergent Phe113 residue of parasite mACP. (**A**) X-ray crystallographic structure of human Isd11 bound to *E. coli* ACP highlighting the central role of the acyl-Ppant group of ACP and R6 residue of Isd11 in stabilizing this complex (PDB entry 5USR). (**B**) Rosetta-based, energy-minimized structural model of the *P. falciparum* mACP-Isd11 interface. Conserved electrostatic interactions that also contribute to the binding interface are shown in *Figure 3—figure supplement 1*. The Rosetta model is included as *Figure 3—source data 1*. (**C**) Nickel-nitrilotriacetic acid (Ni-NTA) pulldown and WB studies of *E. coli* bacteria recombinantly expressing Δ2–50 mACP-HA$_2$ (WT or F113A mutant) and Isd11-His$_6$ and probed with anti-HA and anti-His$_6$ antibodies. All lanes were loaded with equivalent sample volumes, are from the same blot, and were processed identically but were cropped from non-contiguous lanes. The uncropped WB is shown in *Figure 2—figure supplement 3*. (**D**) Continuous growth assays of synchronous mACP-aptamer/TetR-DOZI parasites episomally expressing WT or F113A mACP-HA$_2$ and grown in the presence or absence of anhydrotetracycline (aTc). Data points and error bars are the average and standard deviation from two biological replicates. The presence of an HA epitope tag on both the endogenous and episomal mACP prevented determination of endogenous mACP expression differences ± aTc by WB. WBs are representative of 2–3 replicate experiments with independent samples.

The online version of this article includes the following figure supplement(s) for figure 3:

**Source data 1.** PDB file of energy-minimized Rosetta model of the *P. falciparum* Isd11-mACP complex.

**Figure supplement 1.** Conserved electrostatic interactions at the mACP-Isd11 interface.

**Figure supplement 2.** Uncropped image of western blot (WB) probing interaction of WT or F113A mACP-HA$_2$ and Isd11-His$_6$ in *E. coli*.

mACP (*Figure 3—figure supplement 1*; *Cory et al., 2017*). As an initial step to understand the divergent molecular features of the *P. falciparum* mACP-Isd11 interface that stabilize complexation in the absence of an acyl-Ppant group, we generated a homology model of this complex using published structures and then carried out energy optimization and refinement using the Rosetta software (*Park et al., 2018*). The resulting low-energy structural model suggested that the unusual Phe113 residue of mACP partitions into a hydrophobic pocket at the mACP-Isd11 interface composed of the aliphatic side-chain methylene groups of Lys10 and Arg14 and the phenyl ring of Tyr13 of Isd11 as well as Val117 on mACP (*Figure 3B*). The predicted binding interface also included conserved electrostatic interactions between basic residues on parasite Isd11 and acidic groups on mACP (*Figure 3—figure supplement 1*).

To test if the divergent Phe113 residue of mACP contributes substantially to its association with Isd11, we heterologously coexpressed Isd11-His$_6$ and Δ2–50 mACP-HA$_2$ (WT or the F113A mutant) in *E. coli* and observed robust expression of all recombinant proteins in bacterial lysates. We then used Ni-NTA to selectively pull down His-tagged Isd11 and observed that WT mACP-HA$_2$ but not the F113A mutant co-purified with Isd11 (*Figure 3C*, *Figure 3—figure supplement 2*). These results indicate that mutation of Phe113 to Ala substantially weakens the mACP-Isd11 interaction, supporting the conclusion that Phe113 contributes to the mACP-Isd11 binding interface in the mACP-Isd11-Nfs1 complex as suggested by structural modeling (*Figure 3B*).

To test the functional impact of the Phe113Ala mACP mutation on parasite growth, we transfected our mACP knockdown parasite line with episomes encoding either WT or F113A mACP. Parasites expressing a second copy of WT mACP showed identical growth in the presence or absence of aTc. In contrast, parasites that episomally expressed the F113A mACP mutant showed strongly diminished growth upon aTc removal and loss of endogenous mACP (*Figure 3D*). These results support the conclusion that the Phe residue of mACP plays a key role in stabilizing the mACP-Isd11 association required for Nfs1 stability and function.

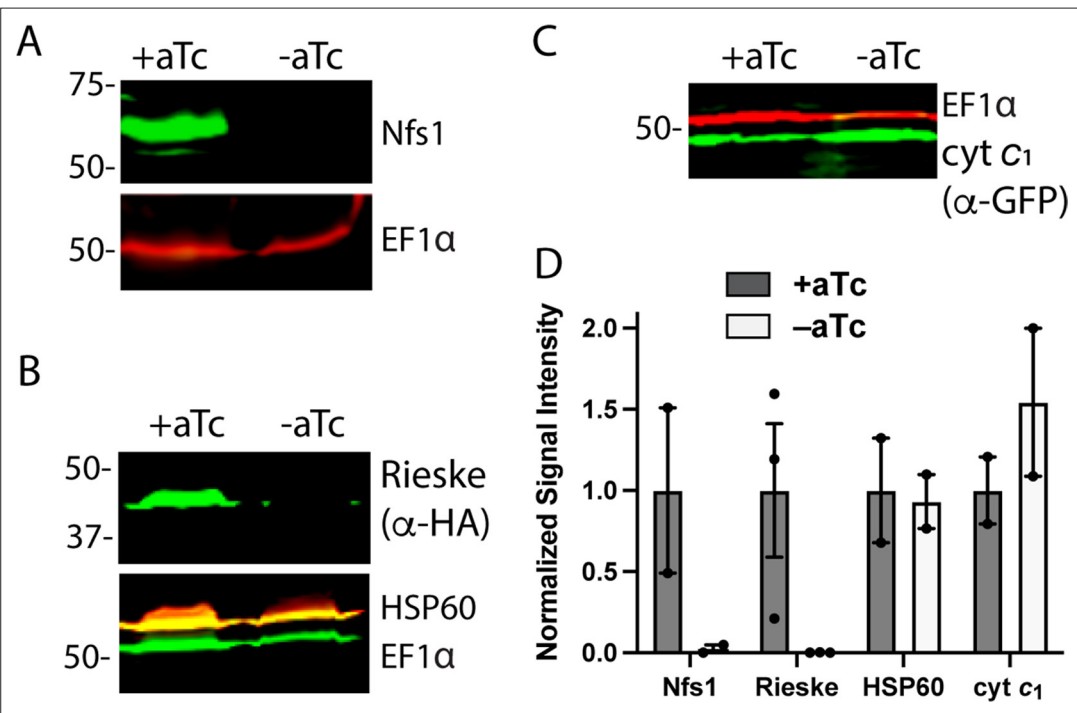

**Figure 4.** Loss of mACP specifically destabilizes Nfs1 and the Rieske Fe-S protein. Western blot analysis of equivalent volumes of lysates from mACP-aptamer/TetR-DOZI Dd2 parasites without or with episomal expression of Rieske-HA$_2$ or cyt $c_1$-GFP that were grown ±aTc for 3 days and probed for cytosolic EF1α (~50 kDa) and (**A**) Nfs1, (**B**) HA-tag and HSP60, or or (**C**) GFP. For (**B**), the blot was first probed with rat anti-HA and rabbit anti-HSP60 1° antibodies, imaged after probing with anti-rat and anti-rabbit 2° antibodies, probed additionally with rabbit anti-EF1α antibody, and imaged again after probing with anti-rabbit 2° antibody. The diminished intensity of EF1α in the -aTc relative to +aTc samples likely reflects the slowing growth of these late second-cycle parasites cultured -aTc due to reduced fitness from loss of mACP. (**D**) Mean ± SEM of the integrated signal intensity for each indicated protein under ± aTc conditions relative to the loading control (EF1α or EF1$\beta$) and normalized to +aTc conditions. Black circles are individual data points from 2–3 replicate experiments with independent samples. Normalized signal intensity values for +aTc versus -aTc conditions were analyzed by two-tailed unpaired t-test to determine the following p values: Nfs1 (0.2), Rieske-HA$_2$ (0.07), HSP60 (0.9), and cyt $c_1$ (0.4). Uncropped blots are shown in *Figure 4—source data 1*.

The online version of this article includes the following figure supplement(s) for figure 4:

**Source data 1.** Raw integrated signal intensity values for western blot densitometry analysis.

**Figure supplement 1.** Validation of anti-Nfs1 antibody for specific recognition of the parasite Nfs1 homolog.

## Loss of mACP destabilizes Nfs1 and the Rieske protein in ETC complex III

Nfs1 stability in other eukaryotes depends critically on its association with Isd11 and ACP, with loss of ACP resulting in degradation of Nfs1 and lethal defects in Fe-S cluster biogenesis (*Van Vranken et al., 2016*; *Cory et al., 2017*; *Boniecki et al., 2017*; *Lill and Freibert, 2020*; *Cai et al., 2017*). Loss of Nfs1 upon mACP knockdown would therefore be sufficient to explain mACP essentiality in *P. falciparum*. To test the impact of mACP knockdown on Nfs1 protein levels in parasites, we used a commercial anti-Nfs1 antibody (Abcam 229829) raised against a 250-amino acid region of human Nfs1 that is 55% identical to *P. falciparum* Nfs1 and that selectively recognized recombinant *P. falciparum* Nfs1 expressed in *E. coli* (*Figure 4—figure supplement 1*). We grew our mACP aptamer/TetR-DOZI parasites ± aTc, harvested them at the end of the second intraerythrocytic growth cycle before parasite death, and analyzed expression of Nfs1 by WB using this Nfs1 antibody.

In parasites grown +aTc, the anti-Nfs1 antibody recognized a band at ~60 kDa (*Figure 4A*) that is close to the predicted molecular mass of 63 kDa for this protein. WB analysis of parasites grown in -aTc conditions failed to detect this ~60 kDa band (*Figure 4A*), strongly suggesting that Nfs1 stability is tightly coupled to mACP expression and that loss of mACP results in Nfs1 degradation in parasites as it does in yeast (*Van Vranken et al., 2016*). Loss of Nfs1 upon mACP knockdown suggested that parasite death was likely to involve dysfunction in one or more Fe-S cluster-dependent pathways. We set out to understand which pathway defect(s) might explain parasite death.

The mitochondrial iron-sulfur cluster (ISC) biogenesis pathway supplies Fe-S clusters to various mitochondrial proteins and has been suggested from studies in yeast to produce a key intermediate that is exported out of the organelle and elaborated by cytoplasmic proteins to assemble and deliver Fe-S clusters to client proteins in the cytoplasm and nucleus (*Lill and Freibert, 2020*; *Kispal et al., 1999*; *Sharma et al., 2010*; *Rouault and Tong, 2005*). Multiple Fe-S cluster proteins function within the *P. falciparum* mitochondrion (*Haussig et al., 2014*). Aconitase (PF3D7_1342100) and succinate dehydrogenase (PF3D7_1212800) are dispensable for blood-stage parasite growth (*Ke et al., 2015*). Ferredoxin (PF3D7_1214600) is expected to be essential (*Gomes et al., 2015*), but its main role is to provide electrons for the ISC pathway (*Lill and Freibert, 2020*). Class I fumarate hydratase (PF3D7_0927300), which functions in the citric acid cycle but may also contribute to purine scavenging, was refractory to disruption in *P. falciparum* (*Ke et al., 2015*) but successfully deleted in *P. berghei* in a mouse strain-dependent manner (*Jayaraman et al., 2018*). The Rieske protein (PF3D7_1439400), which is an essential component of ETC complex III (*Zhang et al., 1998*), is the only known mitochondrial Fe-S cluster protein that has an unequivocally essential function apart from the ISC pathway. We therefore focused on understanding the effect of mACP knockdown on the Rieske protein as a prior study in yeast showed that mACP knockdown resulted in loss of Rieske (*Van Vranken et al., 2016*).

To probe the status of Rieske upon mACP knockdown in parasites, we transfected the mACP-aptamer/TetR-DOZI parasites with an episome encoding Rieske-HA$_2$. A band corresponding to the epitope-tagged Rieske was detected ~43 kDa in WB analysis of parasites grown + aTc. This band, however, was undetectable in parasites grown 72 hr in -aTc conditions to downregulate mACP expression (*Figure 4B*). These results indicate that knockdown of mACP and subsequent degradation of Nfs1 are accompanied by loss of the Fe-S cluster-dependent Rieske protein. The impact of mACP knockdown on stability of Nfs1 and Rieske was specific to these proteins as loss of mACP in -aTc conditions did not reduce expression levels of the mitochondrial chaperone HSP60 or the integral ETC complex III component, cyt $c_1$ (*Figure 4B and C*).

## Loss of mACP causes ETC failure and sensitizes parasites to mitochondrial depolarization by proguanil

Since the Rieske protein is an essential electron-transfer component of ETC complex III (*Zhang et al., 1998*), we reasoned that loss of mACP and Rieske likely results in ETC failure. In blood-stage parasites, which rely on glycolysis rather than oxidative phosphorylation for ATP synthesis, the essential function of the mitochondrial ETC is to oxidatively recycle the ubiquinone cofactor used by several dehydrogenases, of which dihydroorotate dehydrogenase (DHOD) is most critical (*Painter et al., 2007*; *Ke et al., 2011*). Prior work has shown that exogenous addition of the soluble ubiquinone analog, decyl-ubiquinone (dQ), rescues parasites from ETC dysfunction caused by the ETC complex III inhibitor,

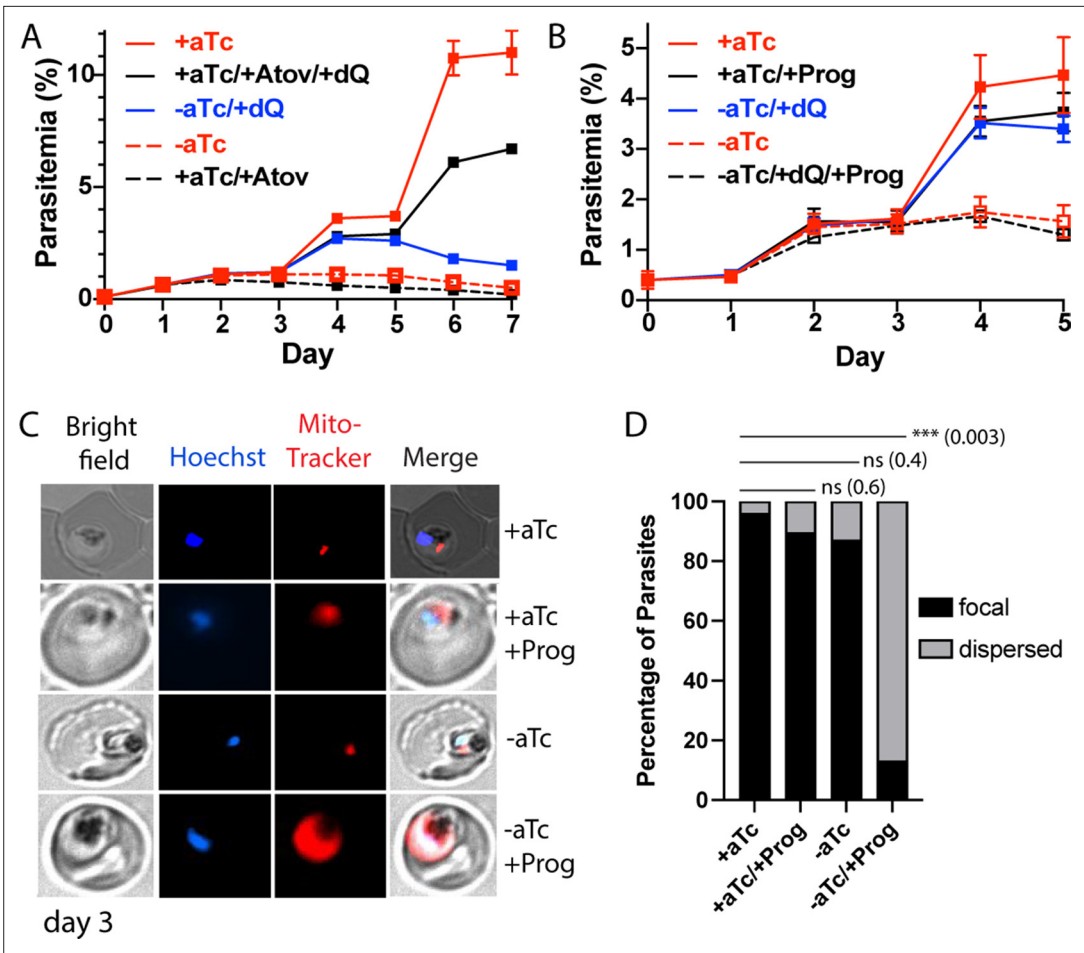

**Figure 5.** Loss of mACP causes electron transport chain (ETC) failure and sensitizes parasites to proguanil. Continuous growth assays of synchronous mACP-aptamer/TetR-DOZI parasites grown (**A**) ± anhydrotetracycline (aTc), ± atovaquone (Atov, 100 nM), ± decyl-ubiquinone (dQ, 15 μM), and (**B**) ± proguanil (Prog, 1 μM). Data points and error bars are the average and standard deviation from 2 to 3 biological replicates. (**C**) Fluorescence microscopy images of live mACP-aptamer/TetR-DOZI Dd2 parasites cultured 3 days ±aTc, 2 days ± 5 μM proguanil, and stained with Hoechst or MitoTracker Red (10 nM). (**D**) Statistical analysis of MitoTracker signal for 40–50 total parasites from each condition in panel (**C**) from two independent experiments. MitoTracker signal was designated as focal if similar in size to nuclear signal (Hoechst stain) or dispersed if similar in size to the parasite cytoplasm (based on brightfield image). For clarity, error bars are not shown but standard errors of the mean were ≤10% in all cases. Cell percentage differences were analyzed by two-tailed unpaired t-test (p values in parentheses, ns = not significant). Parasite counts for individual experiments are given in *Figure 5—source data 1*.

The online version of this article includes the following figure supplement(s) for figure 5:

**Source data 1.** Raw parasite counts for individual microscopy experiments.

**Figure supplement 1.** Additional live microscopy images of mACP-aptamer/TetR-DOZI parasites.

atovaquone (*Ke et al., 2011*). We posited that it might be possible to rescue parasites with dQ from mACP knockdown if the dominant cause of parasite death were ETC failure.

As a positive control and to provide a basis for comparison, we first confirmed that 15 μM exogenous dQ substantially rescued parasites over multiple intraerythrocytic cycles from growth inhibition by 100 nM atovaquone (*Figure 5A*), as previously reported (*Ke et al., 2011*). To test dQ rescue of parasites upon mACP knockdown, we synchronized the mACP-aptamer/TetR-DOZI parasites and monitored their growth ±aTc and ±dQ for several intraerythrocytic cycles. As before, -aTc parasites grew normally for 3 days but failed to expand into the third growth cycle on day 4 (*Figure 5A*). Addition of dQ rescued parasite growth in the third growth cycle in -aTc conditions, similar to dQ rescue of parasites from atovaquone. However, in contrast to indefinite parasite rescue from atovaquone,

dQ only rescued the third-cycle growth of parasites cultured -aTc, which failed to expand further and began to die off on day 6 in the fourth intraerythrocytic cycle (*Figure 5A*). On the basis of dQ rescue, we conclude that ETC failure is the immediate cause of parasite death upon mACP knockdown. However, the inability of dQ to rescue parasites beyond the third cycle indicates that additional dysfunctions beyond the mitochondrial ETC contribute to parasite death on a longer timescale. These additional defects may involve mitochondrial fumarate hydratase and/or essential Fe-S cluster proteins in the cytoplasm and nucleus that depend on Nfs1 function (see 'Discussion' section).

To further unravel the effect of mACP knockdown on ETC function, we next tested if loss of mACP sensitized parasites to mitochondrial depolarization by proguanil. The ETC couples ubiquinone recycling to proton translocation from the mitochondrial matrix to the inner membrane space to establish a transmembrane electrochemical potential. Prior work has shown that parasites retain a second proton-pumping mechanism that also contributes to transmembrane potential and that is inhibited by proguanil (*Painter et al., 2007*; *Skinner-Adams et al., 2019*). Inhibition of ETC function kills parasites due to defective ubiquinone recycling but does not substantially depolarize mitochondria due to activity by this second, proguanil-sensitive pathway. However, ETC dysfunction plus proguanil treatment blocks both pathways for proton pumping and causes mitochondrial depolarization, which can be visualized by a failure to concentrate charged dyes like MitoTracker within the mitochondrion that leads to dispersed dye accumulation in the cytoplasm (*Painter et al., 2007*; *Ke et al., 2018*).

If mACP knockdown and subsequent loss of Rieske caused general ETC failure, we predicted that parasite treatment with sublethal proguanil would negate the ability of dQ to rescue parasite growth from mACP knockdown in the third cycle and would cause mitochondrial depolarization. We repeated the prior growth assay ± aTc and ±dQ but also included 1 µM proguanil. This proguanil concentration alone had no effect on parasite growth, as previously reported (*Painter et al., 2007*). However, when proguanil was combined with growth -aTc, dQ was unable to rescue parasite growth in the third cycle (*Figure 5B*). Microscopy analysis of parasites on day 3 of the growth assay revealed that proguanil treatment selectively prevented mitochondrial accumulation of MitoTracker Red in parasites grown -aTc, strongly suggesting mitochondrial depolarization (*Figure 5C and D*, *Figure 5—figure supplement 1*). This observation strongly supports the model that impaired ubiquinone recycling due to Rieske loss and ETC dysfunction is the immediate cause of parasite death upon mACP knockdown.

## Discussion

The *P. falciparum* mitochondrion is a major antimalarial drug target, but nearly all organelle-specific inhibitors target cytochrome *b* in ETC complex III or DHOD that depends on complex III function (*Antonova-Koch et al., 2018*; *Goodman et al., 2017*). Multiple metabolic pathways operate within the mitochondrion (*van Dooren et al., 2006*), but many are dispensable for blood-stage parasites, and few essential blood-stage functions beyond DHOD and ETC complexes III and IV have been identified. ISC biosynthesis is an ancient, essential mitochondrial function that has been well studied in yeast and mammalian cells but is sparsely studied in parasites (*Gisselberg et al., 2013*). We have identified a divergent and essential nexus between Fe-S cluster biogenesis and an evolutionary vestige of type II fatty acid synthesis in the *P. falciparum* mitochondrion.

### New molecular paradigm for essential ACP function without an acyl-Ppant group

Most eukaryotic cells, including fungi, plants, and animals, retain a mitochondrial FASII pathway in which the acyl-ACP intermediate has been shown to critically mediate respiratory chain assembly, Fe-S cluster biogenesis, and ribosomal translation by binding to LYR-protein assembly factors (*Majmudar et al., 2019*; *Angerer, 2015*; *Nowinski et al., 2018*). These interactions and their dependence on ACP acylation have been proposed to constitute a regulatory feedback mechanism that couples the availability of acetyl-CoA for FASII activity to respiratory chain assembly for oxidative phosphorylation and ATP synthesis (*Van Vranken et al., 2018*). Biochemical coupling of these pathways, however, has not been retained in *Plasmodium*, which has lost mitochondrial FASII enzymes but retains a divergent ACP homolog incapable of 4-Ppant modification or acylation. Nevertheless, we have shown that mACP remains essential to parasites for Fe-S cluster biogenesis by binding to Isd11, which is the only LYR-protein homolog retained by *P. falciparum*. Isd11 binds to mACP via a novel interface molecular

interface, and this interaction is critical stabilizing the mACP-Isd11-Nfs1 cysteine desulfurase complex. *P. falciparum* thus provides a new molecular paradigm for essential ACP function without an acyl-Ppant group. This discovery emphasizes the ancient, fundamental role of ACP in mitochondrial Fe-S cluster biogenesis and suggests an evolutionary driving force to retain mACP interaction with the Isd11-Nfs1 complex-independent ACP's scaffolding role in fatty acid synthesis.

Prior in vitro work has provided evidence that activity of purified Isd11-Nfs1 does not strictly require association with ACP (**Boniecki et al., 2017**; **Lill and Freibert, 2020**). However, Nfs1 is unstable in its native mitochondrial context in the absence of ACP, as shown herein for *P. falciparum* and in prior studies of yeast (**Van Vranken et al., 2016**; **Adam et al., 2006**). Coupling Nfs1 stability (via Isd11) to ACP acylation in eukaryotes that retain mitochondrial FASII has been proposed as a mechanism to up- and downregulate Fe-S cluster biogenesis congruent with nutrient availability and cellular Fe-S cluster needs for respiration and growth (**Lill and Freibert, 2020**). This mechanism, however, cannot explain mitochondrial retention of an acylation-incompetent ACP homolog in *Plasmodium*. ACP may therefore play additional functional and/or regulatory roles that are essential for mitochondrial Fe-S cluster biogenesis and utilization in cells, perhaps involving metal-ion binding and sensing by ACP (e.g., iron or zinc) (**Herrera et al., 2019**; **Qiu and Janson, 2004**) or interactions with other mitochondrial networks akin to broad ACP interactions observed in *E. coli* (**Gully et al., 2003**). In this regard, it is interesting to note that anaerobic eukaryotic parasites like *Giardia* and *Cryptosporidium* lack a mitochondrion but retain a primitive mitosome for Fe-S cluster biogenesis that lacks ACP and Isd11 homologs in this compartment, where Nfs1 appears to function autonomously (**Richards and van der Giezen, 2006**; **Braymer et al., 2021**).

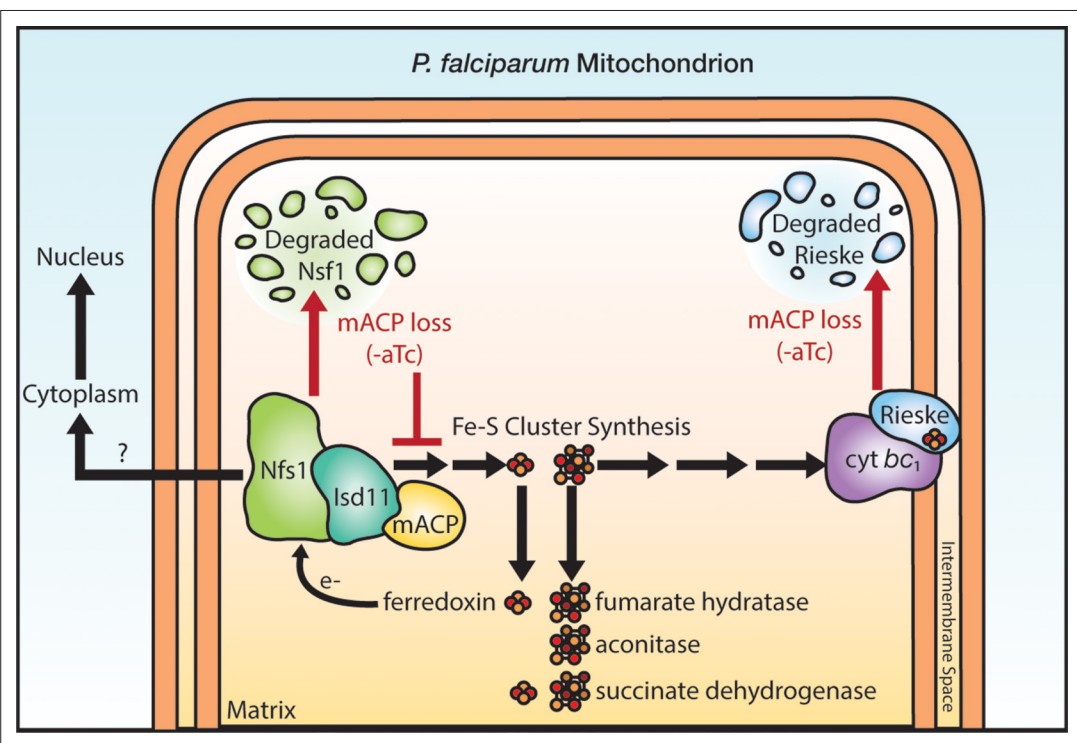

**Figure 6.** Schematic model for the impact of mACP knockdown (-aTc) on Fe-S cluster metabolism in *P. falciparum* parasites. The question mark indicates uncertainty in the functional role of mitochondrial Nfs1 in supporting cytoplasmic Fe-S cluster biogenesis in parasites. For simplicity, the mACP/Isd11/Nfs1 complex is shown with only one monomer for each protein, rather than the functional dimer (**Boniecki et al., 2017**). Illustration by Megan Okada.

The online version of this article includes the following figure supplement(s) for figure 6:

**Figure supplement 1.** Sequence alignment of *P. falciparum* mACP (PF3D7_1208300) with homologs from *T. gondii* (TGME49_270310, e-value 2e-13), *B. microti* (BMR1_02g01455, e-value 7e-13), and *V. brassicaformis* (CEM12831, e-value 3e-16).

## Implications for parasites

Loss of mitochondrial FASII, but retention of a Ppant-independent mACP lacking the conserved Ser, appears to be widespread among apicomplexan parasites, including *Toxoplasma* and *Babesia* (*Figure 6—figure supplement 1*). In the case of *Toxoplasma gondii*, BLAST analysis revealed two divergent mACP homologs (TGME49_270310 and TGME49_265538), differing in the presence of a Cys or Gly in place of the conserved Ser, in their transcriptional profiles, and in their predicted contribution to parasite infection of human fibroblasts (*Sidik et al., 2016*). Outside the phylum Apicomplexa, the only genomic evidence we found for a divergent ACP lacking the conserved Ser was in the related photosynthetic chromerid, *Vitrella brassicaformis*, but not in its algal cousin, *Chromera velia* (*Figure 6—figure supplement 1*). Thus, the adaptation of mitochondrial ACP to function without a Ppant group or acylation likely occurred on a similar evolutionary timeframe as the loss of plastid photosynthesis that accompanied the appearance of apicomplexan parasitism (*Janouškovec et al., 2015*).

Our study indicates that mitochondrial ACP function without an acyl-Ppant group is a parasite-specific adaptation that differentiates apicomplexan parasites from humans. Nevertheless, like yeast and human cells (*Van Vranken et al., 2016*; *Majmudar et al., 2019*), we observed that loss of ACP destabilizes Nfs1 to block Fe-S cluster assembly impairs the stability and function of the key Rieske subunit of ETC complex III and leads to parasite death (*Figure 6*). In yeast and mammalian cells, mACP plays a second role in Rieske maturation by binding to MZM1, an LYR-protein chaperone that stabilizes Rieske upon import into the mitochondrial matrix to receive its Fe-S cluster for subsequent insertion into ETC complex III by the AAA-ATPase, BCS1 (*Majmudar et al., 2019*; *Atkinson et al., 2011*). *P. falciparum*, however, appears to lack an MZM1 homolog (*Figure 2—figure supplement 1*), suggesting that parasites have evolved compensatory mechanisms for Rieske maturation in the absence of this ACP-dependent chaperone. In yeast, loss of mACP and/or the Rieske protein, which is a peripheral subunit inserted late in the assembly of ETC complex III, results in accumulation of a late core assembly intermediate of complex III that contains cyt *b*, cyt $c_1$, and other factors (*Van Vranken et al., 2016*; *Atkinson et al., 2011*). Our observation that cyt $c_1$ levels are retained upon loss of parasite mACP and Rieske in -aTc conditions (*Figure 4*) suggests that this late complex III assembly intermediate likely accumulates in *P. falciparum* similar to yeast. However, complex III assembly has been sparsely studied in *Plasmodium* and remains an important future challenge to understand.

Exogenous dQ rescued parasite growth upon mACP knockdown for one intraerythrocytic cycle but did not rescue growth on a longer timescale. This short-term rescue indicates that respiratory failure is the immediate cause of *P. falciparum* death upon loss of mACP (*Painter et al., 2007*; *Ke et al., 2011*). However, defects in essential processes beyond the mitochondrial ETC contribute to parasite death on longer timescales. These defects may involve mitochondrial fumarate hydratase (*Ke et al., 2015*) and/or essential Fe-S cluster proteins in the cytosol and nucleus that depend on Nfs1 activity (*Figure 6*), including Rli1 required for cytoplasmic ribosome assembly and Rad3 and Pri2 that are essential for nuclear DNA metabolism (*Dellibovi-Ragheb et al., 2013*). In yeast, the mitochondrial transporter Atm1 has been proposed to export a key sulfur-containing intermediate required for cytosolic Fe-S cluster assembly (*Kispal et al., 1999*; *Braymer et al., 2021*). In *P. falciparum*, the Mdr2 transporter (PF3D7_1447900) is 34 % identical to yeast Atm1 and has been proposed to perform an analogous function in cytosolic Fe-S cluster assembly (*van Dooren et al., 2006*). However, Mdr2 appears to be dispensable for blood-stage parasites (*van der Velden et al., 2015*), and its possible mitochondrial-export role in cytoplasmic Fe-S cluster biogenesis remains untested. We also note prior studies in mammalian cells which suggest that cytoplasmic Fe-S cluster synthesis in these cells may be independent of mitochondrial Nfs1 activity (*Maio and Rouault, 2020*; *Kim et al., 2018*). Cytoplasmic Fe-S cluster metabolism in *Plasmodium* is sparsely studied, and its functional dependence on mitochondrial ISC proteins remains a key frontier to further test and understand.

In human cells, recent work has identified an LYR-motif protein (L0R8F8/MIEF1) that mediates interaction between mACP and mitochondrial ribosomes, suggesting a role for mACP in regulating mitochondrial translation (*Brown et al., 2017*; *Rathore et al., 2018*). Malaria parasites lack a homolog of L0R8F8 (*Figure 2—figure supplement 1*), and multiple experimental observations strongly suggest that *P. falciparum* mACP does not regulate mitochondrial translation. Prior work in yeast reported that loss of mitochondrial genome expression strongly reduces the stability of cyt $c_1$, due to loss of mitochondria-encoded cyt *b* and destabilization of ETC complex III (*Zara et al., 2004*). Parasite cyt

$b$ is also encoded on the mitochondrial genome. However, we observed that loss of mACP does not diminish the stability of parasite cyt $c_1$ (**Figure 4C**). It was also recently shown that mitoribosome dysfunction in *P. falciparum* results in mitochondrial depolarization (**Ke et al., 2018**). In contrast to this phenotype, we observed that parasites retained their mitochondrial transmembrane potential upon mACP knockdown in -aTc conditions alone (**Figure 5C**). Collectively, these observations suggest that mACP function does not regulate mitochondrial translation in *P. falciparum*. Nevertheless, mACP may have additional functional interactions beyond the Isd11-Nfs1 complex that also contribute to its essentiality in parasites. Such interactions, however, will diverge from known interactions in yeast and humans that involve conserved LYR-motif proteins, and we have on-going studies to fully understand the functional interactome of mACP in the parasite mitochondrion.

The mitochondrial ETC is essential for malaria parasite viability in all lifecycle stages (**Delves et al., 2012**). The unique molecular features of *P. falciparum* mACP and its interaction with Isd11 that underpin essential functions of the ETC and broader Fe-S cluster utilization suggest the possibility of targeting this complex for antimalarial therapy. We have ongoing structural and biochemical studies to assess if parasite Isd11 retains a vestigial acyl-pantetheine binding pocket and to test if parasite-specific features of the mACP-Isd11 interaction can be selectively disrupted via small-molecule inhibitors that mimic the acyl-pantetheine group and/or broader hydrophobic features of this protein-protein interface that are distinct from the human complex. On the basis of molecular conservation in other apicomplexan parasites, such inhibitors would likely function against other human pathogens such as *Toxoplasma* and *Babesia*.

# Materials and methods

**Key resources table**

| Reagent type (species) or resource | Designation | Source or reference | Identifiers | Additional information |
|---|---|---|---|---|
| Cell line (*Plasmodium falciparum*) | Dd2 | PMID:1970614 | BEI Resources MRA-156 | |
| Cell line (*Plasmodium falciparum*) | Dd2 mACP-HA-FLAG 10X Aptamer/TetR-DOZI | Made for this study | | Described in Materials and methods. Can be obtained from Sigala lab. |
| Cell line (*Plasmodium falciparum*) | Dd2 Cyt $c_1$-GFP (pTEOE) mACP-HA-FLAG 10X Aptamer/TetR-DOZI | Made for this study | | Described in Materials and methods. Can be obtained from Sigala lab. |
| Cell line (*Plasmodium falciparum*) | Dd2 Rieske-HA$_2$ (pTEOE) mACP-HA-FLAG 10X Aptamer/TetR-DOZI | Made for this study | | Described in Materials and methods. Can be obtained from Sigala lab. |
| Cell line (*Plasmodium falciparum*) | Dd2 mACP-HA$_2$ (pTEOE) mACP-HA-FLAG 10X Aptamer/TetR-DOZI | Made for this study | | Described in Materials and methods. Can be obtained from Sigala lab. |
| Cell line (*Plasmodium falciparum*) | Dd2 F113A mACP-HA$_2$ (pTEOE) mACP-HA-FLAG 10X Aptamer/TetR-DOZI | Made for this study | | Described in Materials and methods. Can be obtained from Sigala lab. |
| Cell line (*Plasmodium falciparum*) | Dd2 mACP-HA$_2$ (pTEOE) | Made for this study | | Described in Materials and methods. Can be obtained from Sigala lab. |
| Cell line (*Plasmodium falciparum*) | Dd2 aACP-HA$_2$ (pTEOE) | Made for this study | | Described in Materials and methods. Can be obtained from Sigala lab. |
| Cell line (*Plasmodium falciparum*) | Dd2 mACP-HA$_2$ (pTEOE) Isd11-GFP (pRL2) | Made for this study | | Described in Materials and methods. Can be obtained from Sigala lab. |
| Cell line (*Escherichia coli*) | BL21/DE3 mACP-HA$_2$ (pET28a) | Made for this study | | Described in Materials and methods. Can be obtained from Sigala lab. |

*Continued on next page*

*Continued*

| Reagent type (species) or resource | Designation | Source or reference | Identifiers | Additional information |
|---|---|---|---|---|
| Cell line (*Escherichia coli*) | BL21/DE3 $\Delta$2–51 mACP-HA$_2$ (pET28a) | Made for this study | | Described in Materials and methods. Can be obtained from Sigala lab. |
| Cell line (*Escherichia coli*) | BL21/DE3 Isd11-His$_6$ (pET21d) mACP-HA$_2$ (pET28a) | Made for this study | | Described in Materials and methods. Can be obtained from Sigala lab. |
| Cell line (*Escherichia coli*) | BL21/DE3 LYR12-14AAA Isd11-His$_6$ (pET21d) mACP-HA$_2$ (pET28a) | Made for this study | | Described in Materials and methods. Can be obtained from Sigala lab. |
| Cell line (*Escherichia coli*) | BL21/DE3 Isd11-His$_6$ (pET21d) $\Delta$2–51 mACP-HA$_2$ (pET28a) | Made for this study | | Described in Materials and methods. Can be obtained from Sigala lab. |
| Cell line (*Escherichia coli*) | BL21/DE3 Isd11-His$_6$ (pET21d) F113A ($\Delta$2–51) mACP-HA$_2$ (pET28a) | Made for this study | | Described in Materials and methods. Can be obtained from Sigala lab. |
| Cell line (*Escherichia coli*) | BL21/DE3 Isd11-His$_6$ (pET21d) $\Delta$2–40 aACP-HA$_2$ (pET28a) | Made for this study | | Described in Materials and methods. Can be obtained from Sigala lab. |
| Chemical compound, drug | Anhydrotetracycline | Caymen Chemicals | Cat. no. 10009542 | |
| Chemical compound, drug | Atovaquone | Caymen Chemicals | Cat. no. 95233184 | |
| Chemical compound, drug | Decyl-ubiquinone | Caymen Chemicals | Cat. no. 55486005 | |
| Chemical compound, drug | Proguanil | Sigma-Aldrich | Cat. no. 637,321 | |
| Antibody | Anti-HA (rat monoclonal 3F10) | Sigma-Aldrich | Cat. no. 11867423001, RRID:AB_390918 | (1:1000) |
| Antibody | Anti-HSP60 (rabbit polyclonal) | Novus | Cat. no. NBP2-12734 | (1:1000) |
| Antibody | Anti-Nfs1 (rabbit polyclonal) | Abcam | Cat. no. ab229829 | (1:1000) |
| Antibody | Anti-GFP (goat polyclonal) | Abcam | Cat. no. ab5450, RRID:AB_304897 | (1:1000) |
| Antibody | Anti-EF1$\alpha$ (rabbit polyclonal) | PMID:11251817 | | (1:1000) |
| Antibody | Anti-EF1$\beta$ (rabbit polyclonal) | PMID:11251817 | | (1:1000) |
| Antibody | Anti-His$_6$-DyLight680 (mouse monoclonal) | Thermo Fisher | Cat. no. MA121315D680, RRID:AB_2536987 | (1:1000) |

## Cloning

For episomal protein expression, the genes encoding mitochondrial ACP (PF3D7_1208300), Rieske protein (PF3D7_1439400), cytochrome $c_1$ (PF3D7_ 1462700), and apicoplast ACP (PF3D7_0208500) were PCR-amplified from Dd2 parasite cDNA using primer sets 1/2, 3/4, and 5/6, and 27/28 (*Supplementary file 1*), respectively, and cloned into pTEOE (*Sigala et al., 2015*) at the XhoI/AvrII sites in frame with a C-terminal dual hemagglutinin (HA$_2$) tag (mACP and Rieske) or GFP tag (cytochrome $c_1$). The pTEOE vector contains the HSP86 promoter to drive episomal protein expression, encodes human DHFR as a positive selection cassette (*Sigala et al., 2015*), and is co-transfected with plasmid

pHTH that contains the piggyBac transposase (*Balu et al., 2005*) for integration into the parasite genome. Ligation-independent cloning was performed with the QuantaBio RepliQa HiFi Assembly Mix (VWR 95190-050). Cloning reaction mixes were transformed into Top10 chemically competent cells, and bacterial clones were selected for carbenicillin (Sigma C3416) resistance. Plasmid DNA was isolated using the PureLink Plasmid Miniprep system (Invitrogen K210011), and correct plasmid insert sequences were confirmed by Sanger sequencing (University of Utah DNA Sequencing Core) using vector-specific primers.

For recombinant protein expression in *E. coli*, the gene for *P. falciparum* Nfs1(PF3D7_0727200) was cloned from Dd2 parasite cDNA, and genes for mACP, truncated mACP (Δ2–50), and truncated aACP (Δ2–40) were subcloned by PCR from the pTEOE plasmid using primer pairs 7/8, 9/10, 11/12, and 29/30 (*Supplementary file 1*), respectively. These genes were inserted into the NcoI/XhoI sites of pET28a (Novagen 69864) with a C-terminal $HA_2$ tag using ligation-independent cloning with the QuantaBio system. *P. falciparum* Isd11 was PCR-amplified from Dd2 parasite cDNA using primer sets 13/14 and cloned into the NcoI/XhoI sites of pET21d (Novagen/MilliporeSigma 69743) in-frame with the C-terminal $His_6$ tag using ligation-independent methods. Correct insert sequences were verified by Sanger sequencing of purified plasmid DNA.

## CRISPR-Cas9 genome editing

CRISPR/Cas9-stimulated repair by double-crossover homologous recombination was used to tag the mACP gene to encode a C-terminal HA-FLAG epitope fusion tag and the 3′ 10X aptamer/TetR-DOZI system (*Ganesan et al., 2016*) to enable regulated mACP expression using anhydrotetracycline (aTc, Caymen Chemicals 10009542). A guide RNA sequence corresponding to TGGTATTGTTATATTAAATT was cloned with ligation-independent methods using primer pair 15/16 (*Supplementary file 1*) into a modified version of the previously published pAIO CRISPR/Cas9 vector (*Spillman et al., 2017*) in which the BtgZI site was replaced with a unique HindIII site to facilitate cloning. To tag the mACP gene, the donor pMG75 repair plasmid was created by using ligation-independent cloning to insert a gBlock gene fragment ordered from IDT into the unique AscI/AatII cloning sites. This gBlock contained 200 bp of the 3′ untranslated region of the mACP gene (starting at position 135 downstream from the TAA stop codon), an AfeI site, and the 312 bp of the 3′ end of the mACP coding sequence (excluding the 129 bp intron). The gBlock sequence included a shield mutation to ablate the CRISPR PAM sequence CGG that immediately follows the gRNA sequence above in the antisense strand of the coding sequence by mutating it to CTG, resulting in a silent mutation of the Ser125 codon from TCC to TCT. Before transfection, the pMG75 vector was linearized by AfeI digestion performed overnight at 37°C , followed by deactivation with Antarctic Phosphatase (NEB M0289S).

## Site-directed mutagenesis

The mACP Phe113Ala (pTEOE) and Isd11 LYR12-14AAA (pET21d) mutations were introduced by PCR using primer pairs 17/18 and 19/20, respectively. For mACP, primer pairs 9/18 and 17/10 were used to introduce the Phe113Ala mutation and generate two insert fragments by PCR using the mACP-pET28a plasmid as template. The two fragments were joined by overlap sewing PCR using primer set 9/10. The same PCR process and primers pairs 13/20 and 19/14 were used to make Isd11 LYR12-14AAA using Isd11-pET21d as the template. Primer pairs 24/14, 25/14, and 26/14 were used to make the respective single Isd11 LYR mutations of L12A, Y13A, and R14A. Correct insert sequences for the mutated genes were confirmed by Sanger sequencing of plasmid DNA. Primer sequences are listed in *Supplementary file 1*.

## Parasite culturing and transfection

All experiments were performed using *P. falciparum* Dd2 parasites (*Wellems et al., 1990*), whose identity was confirmed based on expected drug resistance. Parasite cultures were mycoplasma-free by PCR test. Parasite culturing was performed in Roswell Park Memorial Institute medium (RPMI-1640, Thermo Fisher 23400021) supplemented with 2.5 g/L Albumax I Lipid-Rich BSA (Thermo Fisher 11020039), 15 mg/L hypoxanthine (Sigma H9636), 110 mg/L sodium pyruvate (Sigma P5280), 1.19 g/L HEPES (Sigma H4034), 2.52 g/L sodium bicarbonate (Sigma S5761), 2 g/L glucose (Sigma G7021), and 10 mg/L gentamicin (Invitrogen Life Technologies 15750060). Cultures were maintained at 2%

hematocrit in human erythrocytes obtained from the University of Utah Hospital blood bank, at 37°C, and 5% $CO_2$.

For episomal protein expression using the pTEOE vector, parasite-infected erythrocytes were transfected in 1× cytomix containing 50–100 µg of purified plasmids and 25 µg of the pHTH transposase plasmid by electroporation in 0.2 cm cuvettes using a Bio-Rad Gene Pulser Xcell system (0.31 kV, 925 µF). Transfected cultures were allowed to expand in the absence of drug for 48 hr and then selected in 5 nM WR99210 (Jacobus Pharmaceuticals). Stable drug-resistant parasites returned from transfection in 2–8 weeks. The pRL2 plasmid encoding Isd11-GFP (*Gisselberg et al., 2013*) was a gift from Sean Prigge (Johns Hopkins University). This plasmid was transfected into stably selected mACP-HA$_2$ (pTEOE) Dd2 parasites and selected with 5 nM WR99210 and 6 µM blasticidin-S (ThermoFisher/Gibco R21001). Similarly, pTEOE plasmids for Rieske-HA$_2$, cyt $c_1$-GFP, and WT or Phe113Ala mACP-HA$_2$ were separately transfected into the polyclonal Dd2 mACP-HA-FLAG aptamer/TetR-DOZI knockdown line (described below) and selected with 5 nM WR99210, 6 µM blasticidin-S, and 0.5 µM aTc.

For CRISPR/Cas9-based editing of the mACP genomic locus, 50–100 µg each of the linearized pMG75 donor plasmid and pAIO plasmid (expressing gRNA-Cas9) were mixed and transfected into Dd2 parasites and selected with blasticidin-S in the presence of 0.5–1 µM aTc. Polyclonal parasites returning from transfection were genotyped by PCR using the primer sets 21/22 and 1/22 (*Supplementary file 1*) for the wild-type mACP gene. Primer sets 21/23 and 1/23 were used to test parasites for integration and tagging of the mACP gene. This analysis indicated that DNA recombination with the 5′ homology arm occurred upstream of the single intron in the WT genomic locus, resulting in the edited gene lacking this intron. A PCR amplicon for the edited but not WT (unmodified) mACP gene was detected in polyclonal parasites, which were used for most subsequent experiments. Clonal parasite integrants were isolated by limiting dilution of the polyclonal culture and were PCR-genotyped using the same primer sets as above. All growth assays involving conditional mACP expression were performed with polyclonal parasites except for dQ rescue experiments in *Figure 5*, which were performed with clone B5.

## Parasite growth assays

Parasites were synchronized to rings by treatment with 5% D-sorbitol. For growth assays involving regulated mACP expression, aptamer-tagged parasites were washed three times after synchronization with RPMI media lacking aTc and then divided into two equal parts before supplementing one part with 0.5 µM aTc. Growth was monitored by diluting sorbitol-synchronized parasites to ~0.5% parasitemia and allowing culture expansion over several days with daily media changes. For dQ (Caymen Chemicals 55486005) rescue experiments, dQ was dissolved in DMSO and added directly to cultures with a final concentration of 15 µM and ≤0.3% DMSO. Proguanil (Sigma 637321) was added to cultures at a final concentration of 1 µM at the beginning of growth experiments (*Figure 5B*) and added at 5 µM at the end of the first cycle (24 hr after synchronization) in mitochondrial depolarization studies (*Figure 5C*). Atovaquone (Caymen Chemicals 95233184) was used as a positive control for dQ-rescue and membrane-polarization experiments and was dissolved in DMSO and added to corresponding cultures at a final concentration of 100 nM and ≤0.3% DMSO. Parasitemia was monitored daily by flow cytometry by diluting 10 µL of each parasite culture well into 200 µL of 1.0 µg/mL acridine orange (Invitrogen Life Technologies A3568) in phosphate buffered saline (PBS) and analysis on a BD FACSCelesta system monitoring SSC-A, FSC-A, PE-A, FITC-A, and PerCP-Cy5-5-A channels. Parasitemia was determined by flow cytometry in ≥2 biological replicates (distinct parasite samples set up in parallel) and reported as an average and standard deviation, as indicated in each figure legend. Graphs were plotted using GraphPad Prism 9.0.

## Immunoprecipitation experiments

Dd2 parasites expressing mACP-HA$_2$ or endogenously tagged mACP-HA/FLAG were harvested by centrifugation, treated with 0.05% saponin (Sigma 84510) in PBS for 5 min at room temperature to lyse erythrocytes, and spun down by centrifugation at 4000 rpm for 30 min at 4°C. For HA-tagged proteins, IP was performed using Pierce anti-HA magnetic beads (Thermo Scientific 88836). Parasite pellets from ~50 mL cultures were lysed in 1 mL 1% Triton (Sigma 9002931) or 1% digitonin in cold 1× PBS plus protease inhibitor (Thermo Scientific A32955). Pellets were dispersed by brief sonication on a Branson sonicator equipped with a microtip probe and then incubated at 4°C for 1 hr on a rotator.

Lysates were clarified by centrifugation at 13,000 rpm for 10 min. 30 µL of anti-HA magnetic beads was equilibrated in 170 µL cold 1× Tris-buffered saline (20 mM Tris, pH 7.6, 150 mM NaCl) + 0.05% Tween-20 (TBS-T), and beads were collected against a magnetic stand. Parasite lysates supernatants (1 mL) were added to the equilibrated beads and incubated for 1 hr at 4°C by rotation. Beads were washed three times with ice-cold 1× TBS-T. Bound proteins were eluted with ~100 µL 8 M urea (in 100 mM Tris at pH 8.8) and stored at –20°C until use.

Parasite samples expressing GFP-tagged proteins (Isd11-GFP and cyt $c_1$-GFP) were harvested and lysed as above. IP of GFP-tagged proteins was performed using 50 µL of protein A or G Dynabeads (Thermo Scientific 10001D or 10003D) that were equilibrated as described above. 150 µL of cold 1× TBS-T plus 0.5 µL of goat anti-GFP primary antibody (Abcam ab5450) was added to the equilibrated beads and incubated for 10 min before adding the clarified lysate as described above. Washes and elutions were performed as described above.

For MS experiments, IP eluates were precipitated by adding 100% trichloroacetic acid (Sigma 76039) to a final concentration of 20% and incubated on ice for 1 hr in 1.6 mL Eppendorf tubes. Tubes were spun at 13,000 rpm for 25 min at 4°C. Supernatants were removed by vacuum aspiration, and protein pellets were washed once with 500 µL of cold acetone. The protein pellets were air-dried for 30 min and stored at –20°C.

## Western blot analyses

Samples were fractionated by SDS-PAGE using 10% acrylamide gels run at 120 V in the Bio-Rad mini-PROTEAN electrophoresis system. For SDS-PAGE analysis of whole parasite extracts, 1× sample buffer containing beta-mercaptoethanol was added to parasite samples before heating at 95°C for 10 min and centrifuging at 13,000 rpm for 5 min. Fractionated proteins were transferred from polyacrylamide gel to nitrocellulose membrane at 100 V for 1 hr using the Bio-Rad wet-transfer system. Membranes were blocked in 1% casein/PBS for 1 hr at room temperature and then probed with primary antibody overnight at 4°C and secondary antibody at room temperature for 1 hr. Samples containing HA-tagged proteins were probed with a 1:1000 dilution of Roche rat anti-HA monoclonal 3F10 primary (Sigma 11867423001) and a 1:5000 dilution of donkey anti-rat DyLight800 (Invitrogen Life Technologies SA5-10032). Membranes containing GFP-tagged proteins were probed with a 1:1000 of goat anti-GFP polyclonal antibody (Abcam ab5450) and a 1:5000 dilution of donkey anti-rabbit DyLight680 (Invitrogen SA5-10042). Membranes were probed for *P. falciparum* Nfs1 with a 1:1000 dilution of rabbit anti-Nfs1 (Abcam ab229829) 1:5000 dilution of donkey anti-rabbit DyLight800 (Invitrogen SA5-10044). Rabbit antibodies that recognize *P. falciparum* HSP60 (Novus NBP2-12734) or elongation factor alpha or beta (EF1α or EF1β) (*Mamoun and Goldberg, 2001*) were used as loading controls at 1:1000 dilution. Membranes were probed with mouse monoclonal anti-His$_6$-DyLight680 (Invitrogen MA121315D680) at 1:1000 dilution. Membranes were imaged using the Licor Odyssey system. All image adjustments, including contrast and brightness, were linear. WBs were repeated 2–3 times with independent samples. Uncropped images for all WBs are shown in source data.

WBs involving mACP knockdown were analyzed by densitometry using ImageJ (version 2.1.0/1.53c). The integrated signal intensity for each protein of interest (Nfs1, Rieske-HA$_2$, HSP60, or cyt $c_1$-GFP) was normalized to that of the loading control (cytosolic EF1α or EF1β) in the same vertical lane for each experiment. The relative signal intensities (protein of interest/loading control) from 2 to 3 replicate experiments were used to calculate a mean ± SEM and normalized to +aTc conditions for each protein. Differences in normalized signal intensities ±aTc were analyzed by two-tailed unpaired t-test in GraphPad Prism 9.0.

## Mass spectrometry of parasite IP samples

For identification of mACP-interacting proteins in IP experiments, protein samples were reduced and alkylated using 5 mM Tris (2-carboxyethyl) phosphine and 10 mM iodoacetamide, respectively, and then enzymatically digested by sequential addition of trypsin and lys-C proteases, as previously described (*Florens et al., 2006*; *Wohlschlegel, 2009*). The digested peptides were desalted using Pierce C18 tips (Thermo Fisher Scientific), dried, and resuspended in 5% formic acid. Approximately 1 µg of digested peptides was loaded onto a 25-cm-long, 75 µm inner diameter fused silica capillary packed in-house with bulk C18 reversed phase resin (1.9 µm, 100 A pores, Dr. Maisch GmbH). The 140 min water-acetonitrile gradient was delivered using a Dionex Ultimate 3000 ultra high-performance liquid

chromatography system (Thermo Fisher Scientific) at a flow rate of 200 nL/min (Buffer A: water with 3% DMSO and 0.1% formic acid; Buffer B: acetonitrile with 3% DMSO and 0.1% formic acid). Eluted peptides were ionized by the application of distal 2.2 kV and introduced into the Orbitrap Fusion Lumos mass spectrometer (Thermo Fisher Scientific) and analyzed by tandem mass spectrometry. Data was acquired using a Data-Dependent Acquisition method consisting of a full MS1 scan (resolution = 120,000) followed by sequential MS2 scans (resolution = 15,000) to utilize the remainder of the 3 s cycle time. Data analysis was accomplished using the Integrated Proteomics Pipeline 2 (Integrated Proteomics Applications, San Diego, CA). Data was searched against the protein database from *P. falciparum* 3D7 downloaded from UniprotKB (10,826 entries) in October 2013. Tandem MS spectra searched using the ProLuCID algorithm followed by filtering of peptide-to-spectrum matches by DTASelect using a decoy database-estimated false discovery rate of <1%.

To analyze specific enrichment of Nfs1 in the mACP-HA$_2$ IP versus aACP-HA$_2$ IP for each matched experiment, the spectral counts observed for each protein in the mACP IP were divided by the spectral counts observed for the same protein in the aACP IP. For purposes of this analysis, all zero values for spectral counts were arbitrarily converted to a 1. The log$_2$ value of the mACP/aACP spectral count ratio for each protein was plotted for each of the three matched experiments. Nfs1 was highly enriched in the mACP data set and among the top 0.5–1% of proteins based on spectral count ratio.

## Mass spectrometry of purified recombinant protein samples

For experiments involving Ni-NTA pulldown of recombinant *P. falciparum* Isd1l-His$_6$ from *E. coli* coexpressing *P. falciparum* (Δ2–50) mACP (described below), purified proteins were identified by proteolytic digestion and tandem mass spectrometry. Proteins were reduced with DTT for 45 min at 60°C and then alkylated with iodoacetamide for 30 min at room temperature. Proteins were digested overnight at 38°C with Trypsin/LysC mixture using 1 µg of trypsin per sample and quenched by acidification with 1% formic acid to a pH of 2–3. Reversed-phase nano-LC/MS/MS was performed on an UltiMate 3000 RSLCnano system (Dionex) coupled to a ThermoScientific QExactive-HF mass spectrometer equipped with a nanoelectrospray source. Concentrated samples were diluted with a 1:1 ratio of sample:0.1% formic acid in water. 5 µL of the samples were injected onto the liquid chromatograph. A gradient of reversed-phase buffers (Buffer A: 0.2% formic acid in water; Buffer B: 0.2% formic acid in acetonitrile) at a flow rate of 150 µL/min at 60°C was setup. The LC run lasted for 83 min with a starting concentration of 5% buffer B increasing to 55% over the initial 53 min and a further increase in concentration to 95% over 63 min. A 40-cm long/100 µm inner diameter nanocolumn was employed for chromatographic separation. The column is a reverse-phase BEH C18 3.0 µm nanocolumn. MS/MS data was acquired using an auto-MS/MS method selecting the most abundant precursor ions for fragmentation. The mass-to-charge range was set to 350–1800. Mascot generic format (MGF) files were generated from the raw MS/MS data. Mascot (version 2.6) uses the MGF file for database searching and protein identification. For these samples, the Custom database was searched with the *Plasmodium* taxonomy selected. The parameters used for the Mascot searches were trypsin digest; two missed cleavages; carbamidomethylation of cysteine set as fixed modification; oxidation of methionine and acetylation of the n-terminus were set as variable modifications; and the maximum allowed mass deviation was set at 11 ppm.

## Sequence homology searches

Sequence similarity searches of the *P. falciparum* genome for protein homologs of known LYR proteins from yeast and humans (*Angerer, 2015*) were performed by BLASTP analysis as implemented at the Plasmodium Genomics Resource webpage (https://www.plasmodb.org, release 48). The amino acid sequence of each human LYR protein homolog was used as bait, based on the following Uniprot accession codes: LYRM1 (043325), LYRM2 (Q9NU23), LYRM3 (Q9Y6M9), LYRM4 (Q9HD34), LYRM5 (Q6IPRI), LYRM6 (P56556), LYRM7 (Q5U5X 0), LYRM8 (A6NFY7), LYRM9 (A8MSI8), ACN9 (Q9NRP4), C7orf55 (Q96HJ9), and L0R8F8. Only protein hits with e-values <0.01 were retained.

To identify divergent ACP homologs in other organisms that also lacked the conserved Ser, the amino acid sequence for *P. falciparum* mACP starting with Leu51 was submitted as query to NCBI BLASTP (e-value threshold of 0.05). This search identified multiple homologs within the phylum *Apicomplexa* that also lacked the conserved Ser (e.g., *T. gondii* TGME49_270310 and *Babesia microti* BMR1_02g0145). To identify homologs outside of Apicomplexa, this search was repeated with

Apicomplexa as an exclusion criterion. A single protein sequence from *V. brassicaformis* (CEM21745, e-value of 3e-16) was the only ACP homolog identified in this expanded search that lacked the conserved Ser. An alignment of these homologs with *P. falciparum* mACP is shown in *Figure 6—figure supplement 1*.

## Fluorescence microscopy

For live-cell experiments, parasite nuclei were visualized by incubating samples with 1–2 µg/mL Hoechst 33342 (Thermo Scientific Pierce 62249) for 10–20 min at room temperature. The parasite mitochondrion was visualized by incubating parasites with 10 nM MitoTracker Red CMXROS (Invitrogen Life Technologies M7512) for 15 min prior to washout and imaging. 40–50 total parasites for each condition in two independent experiments were scored for focal or dispersed MitoTracker signal, and cell percentages were analyzed by two-tailed unpaired t-test in GraphPad Prism. For immunofluorescence assay (IFA) experiments, parasites were fixed, stained, and mounted, as previously described (*Sigala et al., 2015*; *Tonkin et al., 2004*). For IFA studies, the parasite mitochondrion was visualized using a polyclonal rabbit anti-HSP60 antibody (Novus NBP2-12734) and AlexaFluor 647-conjugated goat anti-rabbit 2° antibody (Invitrogen Life Technologies A21244), the nucleus was stained with ProLong Gold Antifade Mountant with DAPI (Invitrogen Life Technologies P36931), and mACP-HA$_2$ was visualized with a Roche rat anti-HA monoclonal 3F10 primary antibody and FITC-conjugated donkey anti-rat 2° antibody (Invitrogen Life Technologies A18746). Images were taken on DIC/brightfield, DAPI, GFP, and RFP channels using an EVOS M5000 imaging system or Zeiss 880 Laser-Scanning Confocal Microscope fitted with an Airyscan detector. Fiji/ImageJ was used to process and analyze images. All image adjustments, including contrast and brightness, were made on a linear scale.

## Rosetta modeling of the mACP-Isd11 interface

A homology model of *P. falciparum* mACP bound to Isd11 was created using the InterPred modeling interface (*Mirabello and Wallner, 2017*) and the structure of bovine complex I (*Zhu et al., 2016*) that contained mACP bound to mammalian LYR proteins NDUFA6 and NDUFB9 (PDB 5LDW, chains T and W) as template. The homology models for ACP and Isd11 were then superimposed on the previously determined structure (*Cory et al., 2017*) of *E. coli* ACP bound to human Isd11 (PDB 5USR, chains L and D). The structural model for the parasite mACP-Isd11 complex was refined using Rosetta Dual Relax (*Conway et al., 2014*), which repeatedly alternates between coordinate minimization and side-chain packing 20 times with gradually increasing van der Waals repulsion strength. Four different modeling strategies were investigated that varied the template chains in 5USR (L and D or J and H) and whether or not to restrain the refinement on the template coordinates. Ten independent refinements were performed for each modeling strategy and evaluated for Rosetta energy (*Park et al., 2016*). The unrestrained refinement of the starting model using chains L and D resulted in the final model of lowest calculated energy, and this model was used for further analysis. MacPyMOL (Schrodinger) version 1.8 was used for structural visualization.

## Recombinant protein expression in *E. coli* and purification

Chemically competent *E. coli* BL21/DE3 cells were transformed alone or in combination with mACP-HA$_2$/pET28a (full-length or the Δ2–50 truncation and WT or F113A mutant), (Δ2–40) aACP-HA$_2$/pET28a, Isd11-His$_6$/pET21d (WT, YR/AA, or LYR/AAA mutants), or full-length Nfs1-HA$_2$/pET28a plasmids by heat shock. The bacteria were grown at 37°C in 20 mL LB media in the presence of ampicillin (100 µg/mL) or kanamycin (50 µg/mL). Bacterial cultures were allowed to grow to an optical density (at 600 nm) of 0.4–0.6 before inducing protein expression with isopropyl 1-thio-β-galactopyranoside (IPTG) (Goldbio 367931) at a final concentration of 1 mM. Induced cultures were grown overnight at 20°C before harvesting by centrifugation. Bacterial pellets were either used immediately or stored at –20°C . Bacterial pellets were resuspended in 1 mL 1× cold PBS and lysed by sonication on ice using a Branson sonicator equipped with a microtip for five sets of 10 pulses at 50% power and 50% duty cycle. Supernatants were clarified by centrifugation at 13,000 rpm for 10 min. For purifying hexa-His-tagged proteins, 100 µL of Ni-NTA resin (Thermo Scientific 88221) was equilibrated with 1 mL of buffer A (50 mM NaH$_2$PO$_4$.H$_2$0, 500 mM NaCl, and 5 mM imidazole, pH 8.0). 100 µL of the clarified lysate supernatant was added to the equilibrated resin, diluted to 1 mL with buffer A, and incubated for 30 min. Resin was collected by centrifugation and washed three times with buffer A

and once with wash buffer (Buffer A plus 50 mM imidazole). Bound protein was eluted by incubating resin with 100 µL elution buffer B (Buffer A plus 500 mM imidazole) for 15 min on ice. To purify Isd11-containing complexes in lysates from bacteria coexpressing *P. falciparum* His$_6$-Isd11 and $\Delta$2–50 mACP, the imidazole eluate was run over an S-100 size-exclusion column (Cytiva Life Sciences 17116501) on an AKTA FPLC system (Cytiva Life Sciences). Lysates or eluates were diluted into SDS sample buffer and analyzed by SDS-PAGE and WB analysis.

### Anti-Nfs1 antibody validation

The commercial rabbit anti-Nfs1 primary antibody (Abcam ab229829) was reported to be raised against the amino acids 208–457 of human Nfs1, which is 55% identical to *P. falciparum* Nfs1 when aligned by BLAST. To determine if this commercial antibody selectively recognized *P. falciparum* Nfs1, we recombinantly expressed full-length *P. falciparum* Nfs1 or mACP in *E. coli* using the Nfs1-pET28a or mACP-pET28a constructs with a C-terminal dual HA$_2$ tag (described above) that results in protein expression with expected sizes of 65 kDa (Nfs1) or 21 kDa (mACP). Protein expression was induced with IPTG, and bacteria were harvested by centrifugation and lysed in PBS by sonication. Bacterial lysate supernatants were fractionated by 10% SDS-PAGE by diluting 3 µL of clarified lysate into 47 µL of 1× SDS sample buffer, denaturing at 95°C for 10 min, spinning at 13,000 rpm for 5 min to clarify, and loading samples onto a 10% SDS-PAGE gel. After electrophoresis, samples were transferred to nitrocellulose membrane and probed with 1:1000 dilution of Roche Rat anti-HA primary antibody/1:10,000 dilution of goat anti-rat IRDye 680LT secondary antibody (LiCor 92668029) and 1:1000 dilution of rabbit anti-Nfs1 primary antibody /1:10,000 dilution of donkey anti-rabbit IRDye800CW secondary antibody (LiCor 92632213). Detection of a single band at 65 kDa by both anti-HA and anti-Nfs1 antibodies confirmed the ability of these antibodies to recognize the HA-tagged *P. falciparum* Nfs1. The anti-Nfs1 antibody did not recognize recombinant mACP-HA$_2$, confirming its specificity, but did recognize an endogenous *E. coli* protein ~48 kDa that we identified by mass spectrometry as bacterial IscS, which is 60% identical to human Nfs1.

## Acknowledgements

We thank Greg Ducker, Chris Hill, Roland Lill, Sara Nowinski, Jared Rutter, Dennis Winge, and members of the Sigala lab for helpful discussions and Sean Prigge for the plasmid encoding *P. falciparum* Isd11-GFP. We thank Rebecca Marvin for assistance with making the *P. falciparum* cytochrome $c_1$-GFP expression plasmid, Megan Okada for assistance with graphical figure schemes and the design of integration vectors, and Sandra Osburn for assistance with recombinant protein mass spectrometry. PAS holds a Career Award at the Scientific Interface from the Burroughs Wellcome Fund and a Pew Biomedical Scholarship from the Pew Charitable Trusts. SF was supported by the African American Doctoral Scholars Initiative at the University of Utah. Microscopy, flow cytometry, protein mass spectrometry, and DNA synthesis and sequencing were performed using core facilities at the University of Utah.

## Additional information

### Funding

| Funder | Grant reference number | Author |
| --- | --- | --- |
| National Institute of General Medical Sciences | R35GM133764 | Paul A Sigala |
| National Institute of General Medical Sciences | R01GM089778 | James A Wohlschlegel |
| National Institute of Diabetes and Digestive and Kidney Diseases | U54DK110858 | Paul A Sigala |
| Burroughs Wellcome Fund | 1011969 | Paul A Sigala |
| Pew Charitable Trusts | 32099 | Paul A Sigala |

| Funder | Grant reference number | Author |
|---|---|---|
| National Institute of General Medical Sciences | T32GM122740 | Jaime Sepulveda |
| National Institutes of Health | S10OD018210 | Paul A Sigala |

The funders had no role in study design, data collection and interpretation, or the decision to submit the work for publication.

## Author contributions

Seyi Falekun, Conceptualization, Formal analysis, Investigation, Methodology, Validation, Visualization, Writing - original draft, Writing - review and editing; Jaime Sepulveda, Investigation, Methodology, Writing - review and editing; Yasaman Jami-Alahmadi, Hahnbeom Park, Formal analysis, Investigation, Writing - review and editing; James A Wohlschlegel, Funding acquisition, Project administration, Supervision; Paul A Sigala, Conceptualization, Formal analysis, Funding acquisition, Project administration, Supervision, Validation, Writing - original draft, Writing - review and editing

## Author ORCIDs

Seyi Falekun http://orcid.org/0000-0003-2280-4424
Jaime Sepulveda http://orcid.org/0000-0002-3557-4093
Paul A Sigala http://orcid.org/0000-0002-3464-3042

## Decision letter and Author response

Decision letter https://doi.org/10.7554/71636.sa1
Author response https://doi.org/10.7554/71636.sa2

---

# Additional files

## Supplementary files
• Supplementary file 1. PCR primers used for cloning and genotyping.
• Transparent reporting form

## Data availability

All data reported or analyzed in this manuscript are available and included in the main and supplemental figures and in the source data files.

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
