## [Decision Letter]

**Acceptance summary:**

This study defines the role of a divergent mitochondrial-localized isoform of a FASII acyl carrier protein (mACP) in the malaria parasite, *Plasmodium falciparum*. In contrast to the situation in other eukaryotes, mACP is not involved in fatty acid biosynthesis but is primarily involved in stabilizing proteins involved in mitochondrial Fe-S complex formation. The findings suggest that ACP acquired a role in Fe-S complex formation early in eukaryotic evolution and highlights additional components of the Plasmodium respiratory chain that are important for viability.

**Decision letter after peer review:**

Thank you for submitting your article "Divergent Acyl Carrier Protein Decouples Mitochondrial Fe-S Cluster Biogenesis from Fatty Acid Synthesis in Malaria Parasites" for consideration by *eLife*. Your article has been reviewed by 3 peer reviewers, including Malcolm J McConville as the Reviewing Editor and Reviewer #1. and the evaluation has been overseen by Dominique Soldati-Favre as the Senior Editor. The following individuals involved in review of your submission have agreed to reveal their identity: Lilach Sheiner (Reviewer #2); Geoff McFadden (Reviewer #3).

Essential revisions:

The reviewer's recognized the importance of these findings and the quality of the data overall. However, they have raised the following points which should be addressed and may require additional experimentation and/or discussion in the text.

(1) Inclusion of a control to show that loss of mACP does not lead to loss of other mitochondrial proteins (other than Nsf1, Rieske protein).

(2) Confirmation that Isd11-GFP is localized in mitochondrion (and that the interactions are not an artifact that occurs after cell lysis).

3) The possibility that mACP may be involved in regulating Fe-S synthesis via other mechanisms (such as by regulating mitochondrial protein translation) should be considered and discussed in both the results and Discussion section.

*Reviewer #1 (Recommendations for the authors):*

This study provides fundamental insights into iron-sulfur complex assembly in eukaryotes, as well as the metabolic requirements of *Plasmodium falciparum* and related apicomplexan parasites. The results and supplementary data are clearly presented and the manuscript very well written. The conclusions are well supported by the data. There are no additional comments for improvement.

*Reviewer #2 (Recommendations for the authors):*

Key experiments to address the major weaknesses:

Firm-up the data about Nsf1 and Rieske instability with independent repetitions, statistical analysis of signal reduction and an appropriate control for other mitochondrial proteins.

Further characterisation of the mutant to provide support for specificity of the defect – according to the hypothesis, functions of mitochondrial protein import, division and translation would presumably show defect after mETC defect is observed.

Other suggestions for additions that could enhance rigour and provide support for the conclusions

The reciprocal co-IP with epitomal Isd1-GFP: it is critical to show this fusion is mitochondrial (the interaction could take place in the lystae stage). For the same reason, a negative control for the reciprocal co-IP with aACP (which provided nice support for the specific interaction with the other Fe-S synthesis component, NSf1) would provide further support for the existence of a mitochondrial specific complex between the three.

Nsf1-mACP-Isd11 interaction and Rieske stability: A native gel and western might help in providing support for complex assembly between mACP-Nsf1-Isd11, and its dissociation in the mutant. This could also be used for further support of the mutagenesis work (Phe111Ala) to study the interaction directly in the parasite.

Some information/data is missing: a detailed list of MS outcome from IPs with mACP and aACP (peptides sequences, peptide numbers, proteins identified, scores); The experiment demonstrating lack of interactions of parasite Isd1 with *E. coli* ACP, along with relevant controls, is not shown; the data supporting the phylogenetic discussion in lines 503-507 is not fully shown.

A suggestion for enhancing clarity: Some details are hard to find in the manuscript, and might be missing, or need mentioning in context (such as in the relevant point at the result section and within figure legends). For example: what was the cut-off for BLAST search? What do the author mean by "all proteins" in lines 242 and 310? What individual complexes are separated in line 245? What cellular compartment is EF1a from? How many independent repeats are represented by the different Western gels? Is the processing found for mACP in line with any mitochondrial processing predictions?

*Reviewer #3 (Recommendations for the authors):*

Figure 1B (and Figure 1 supplement 3) depict DAPI staining but only Hoechst staining is described in the Materials and methods (line 607).

Lines 48, 52 and 56 – I'm not a fan of Ppant as an abbreviation for 4-phosphopantetheine. Ppant is not in wide usage (nothing on Google), so why make up your own abbreviations?

Line 60 (and lines 510, 512 and 547)- Some general confusion about high level taxon naming conventions. Apicomplexan is an adjective and not a taxon name, so it would only use a capital 'A' when used at the start of a sentence. It also shouldn't be in italics, which are reserved for species names and not used for higher order taxa. You could say Apicomplexa, in which case you would be referring to the Phylum Apicomplexa (no italics). You can also say apicomplexans (no italics), which is an informal category mostly used as equivalent to Phylum Apicomplexa.

Line 85 – Some readers might take the wrong meaning from this sentence that the ancestral mitochondrial FASII enzymes are now targeted to the apicoplast because of the word 'instead'.

Line 90 – a new abbreviation, different to the one in the abstract, for 4-phosphopantetheine is created (4-Ppant). My recommendation is to lose both abbreviations and spell it out.

Lines 134 and 136- It is molecular mass not weight.

Line 141 – molecular mass not MW.

Line 146 – This reads as though you somehow induced the bacteria to commence the mACP-HA2 protein with a leucine residue rather than a methionine. Reword. Also, here is a good place to introduce the nomenclature (△2-50) used in Figure 1.

Line 165 Figure 1. The DOZI cartoon is perhaps superfluous but if retained, I recommend adding the HA2 tag to the diagram for completeness.

Line 182 – My version of the file doesn't seem to contain the uncropped blot. Also, no need to abbreviate western blotting to WB.

Line 198 – 'three-helical' not 3-helical. Spell out numbers less than 10 unless used in conjunction with a unit of measure.

Line 211 – five and 11 amino acids (see above).

Line 302 – 'stabilizes' not stabilize.

Line 338 (and numerous other places) – no need to abbreviate western blotting to WB.

Line 440 – MitoTracker in the text but Mitotracker in Figure 5C and Figure 5 supplement 1.

Line 462 – italicise 'b' in cytochrome b.

Line 464 – add a comma after 'parasites' at the end of the line.

Line 465 – I would qualify this sentence to refer only to the blood stage as follows. 'and few essential functions in the blood stage beyond DHOD and ETC complexes…..;

Line 503 – insert comma after FASII and another after Ser on line 504

Line 584 – 'samples'.

Line 587 – 'Triton'.

Line 847 Figure 1 supplement 3 – Is the erythrocyte in the first row infected with two parasites? It seems to have two nuclei and two mitochondria. I consider it best to avoid depicting red cells invaded by multiple parasites as it confuses the uninitiated.

Line 651 – The References need lots of fixes. i.e. references 1, 17, 21, 31, 34,35, 36,45,49, 50, 60, and 67 all have extraneous capital letters. References 2, 3, 4, 6, 8, 9, 10, 12, 13, 15, 16, 18, 36, 42, 43, 45, 47, 53, 59, 62, and 63 require the Latin genus and species names to be italicised. References 2, 18, 22, and 63 are missing either page numbers or article identifiers.

Line 834 – Figure 1 supplement 1. This is a nice table (and not a figure by the way), but is it relevant? Various alignments such as Figure 1A and Figure 1 supplement figure 2 provide reasonable confirmation that mACP and aACP are two different acyl carrier proteins, and the assignations of the Fab proteins targeted to the apicoplast are well covered in earlier literature.

The paper uses three letter amino acid abbreviations in the text but single letter amino acid abbreviations in the figures. Consider a consistent system across both.

---

## [Author Response]

Essential revisions:The reviewer's recognized the importance of these findings and the quality of the data overall. However, they have raised the following points which should be addressed and may require additional experimentation and/or discussion in the text.(1) Inclusion of a control to show that loss of mACP does not lead to loss of other mitochondrial proteins (other than Nsf1, Rieske protein).

We have added additional western blot data to Figure 4, including replicates with signal quantitation, which show that levels of the mitochondrial chaperone Hsp60 and the ETC complex III protein cytochrome c_1_ are unaffected by loss of mACP.

(2) Confirmation that Isd11-GFP is localized in mitochondrion (and that the interactions are not an artifact that occurs after cell lysis).

Localization of Isd11-GFP to the *P. falciparum* mitochondrion was previously published by Dr. Sean Prigge’s group in ref. 37, which we cite in the paper. Dr. Prigge sent us his expression plasmid, which we transfected into our mACP-HA_2_ Dd2 line and used for co-IP experiments. We independently confirmed that Isd11-GFP co-localized with MitoTracker Red in the Dd2 line used for our co-IP experiments. We have added this colocalization data as Figure 2—figure supplement 2.

3) The possibility that mACP may be involved in regulating Fe-S synthesis via other mechanisms (such as by regulating mitochondrial protein translation) should be considered and discussed in both the results and Discussion section.

We have modified the Results and Discussion sections to incorporate the perspectives below and acknowledge that mACP may have additional functions and interactions in the *Plasmodium* mitochondrion beyond the Isd11-Nfs1 complex.

Extensive prior studies of mitochondrial ACP in yeast and human cells indicate that mACP function in these organisms (outside of FASII) is mediated by key interactions with LYR motif-containing proteins. Key mACP functions in other eukaryotes are therefore predicted to be mediated by conserved LYR-motif proteins. Our BLAST analysis of the *P. falciparum* proteome identified Isdll as the only LYR-motif protein homolog retained by malaria parasites, strongly suggesting that mACP may have an unusually narrow interactome and thus function in *Plasmodium* relative to yeast and humans. Retention of a conserved interaction between mACP and the Isd11-Nfs1 complex in *Plasmodium* is consistent with the ancient, essential role for Fe-S cluster biogenesis in mitochondria. Indeed, Fe-S cluster biogenesis is arguably the most fundamental function of mitochondria and is retained by eukaryotes that have lost most other mitochondrial functions but retain a vestigial mitosome to support Fe-S cluster synthesis. We fully discuss these concepts in the Results and Discussion sections.

Nevertheless, we acknowledge that mACP may have other functionally relevant interactions in the parasite mitochondrion beyond the Isd11-Nfs1 complex. Such interactions, however, will be divergent (relative to yeast/humans) as they will not involve conserved LYR-motif protein homologs (since Isd11 is the only such parasite homolog we were able to identify). This divergence will make such interactions more challenging to uncover. We have on-going studies to better understand possible roles for mACP in other essential mitochondrial functions (e.g., maturation of the Rieske protein) beyond interaction with Isd11-Nfs1. Nevertheless, we emphasize that the essential role for mACP in stabilizing the Isd11-Nfs1 complex identified in our study is sufficient to explain the observed phenotypes we report in our manuscript and thus account for retention of mACP despite loss of mitochondrial FASII.

With respect to mitochondrial translation, multiple observations strongly suggest that mACP knock-down in *Plasmodium* does not substantially impact translation of the mitochondrial genome:

1. In human cells, mACP was observed to bind a novel LYR-motif protein L0R8F8 that associates with mitochondrial ribosomes and has been proposed to mediate ribosome assembly (references 23 and 63). The functional consequence of this mACP-mito ribosome interaction and its dependence on mACP acylation remain incompletely understood. This interaction does not appear to be generally conserved in all eukaryotes, as yeast do not express a homolog of L0R8F8. We also failed to identify a L0R8F8 homolog in *Plasmodium*, suggesting that parasites lack this interaction between mACP and mito ribosomes.

2. Mitochondrial translation remains poorly studied in *Plasmodium*, due in part to a paucity of validated antibodies that can detect mitochondrial-encoded proteins. Nevertheless, recent work from Dr. Hangjun Ke’s lab (which we cite in ref. 47) reported that knockdown of a mito ribosome protein in *P. falciparum* (expected to lead to defective mitochondrial translation) resulted in mitochondrial depolarization. In contrast to Dr. Ke’s observation, mACP knockdown alone in our study did not result mitochondrial depolarization (Figure 5). These contrasting phenotypes suggest that loss of mACP does not substantially impact mitochondrial translation in parasites.

3. Prior work in yeast (reference 64) has shown that loss of mitochondrial translation results in a substantial ~70% reduction in cyt c_1_, presumably due to loss of cyt *b* expression and destabilization of complex III, of which cyt c_1_ is a core component. In our revised Figure 4, we now show that loss of mACP expression in parasites does not diminish cyt c_1_ levels. These contrasting phenotypes strongly suggest that mACP knockdown in parasites does not impact mitochondrial translation.

Reviewer #2 (Recommendations for the authors):Key experiments to address the major weaknesses:Firm-up the data about Nsf1 and Rieske instability with independent repetitions, statistical analysis of signal reduction and an appropriate control for other mitochondrial proteins.

We have provided biological replicates and signal quantitation and show that levels of mitochondrial Hsp60 and cyt c_1_ are unaffected by mACP knockdown.

Further characterisation of the mutant to provide support for specificity of the defect – according to the hypothesis, functions of mitochondrial protein import, division and translation would presumably show defect after mETC defect is observed.

The data strongly supports our conclusion that mACP knockdown specifically destabilizes Nfs1 and Rieske but not other mitochondrial proteins such as Hsp60 or cyt *c_1_*. We also see no evidence that mACP knockdown inhibits mitochondrial translation or polarization (or protein import and processing that depend on transmembrane potential).

Other suggestions for additions that could enhance rigour and provide support for the conclusions.The reciprocal co-IP with epitomal Isd1-GFP: it is critical to show this fusion is mitochondrial (the interaction could take place in the lystae stage). For the same reason, a negative control for the reciprocal co-IP with aACP (which provided nice support for the specific interaction with the other Fe-S synthesis component, NSf1) would provide further support for the existence of a mitochondrial specific complex between the three.

Please see our response to essential revision #2 above. Mitochondrial localization for Isd11-GFP has previously been published. Our IP-MS data in Figure 2 strongly address the second point and indicate that mACP but not aACP co-IPs with Nfs1. We have added bacterial co-IP data that also show that mACP (Figure 2) but not aACP (Figure 2—figure supplement 5) co-IPs with Isd11 when these proteins are co-expressed in *E. coli*. This specificity is consistent with our discussion of the adaptive sequence features of parasite Isd11 that loss of Arg6 favors association of Isd11 with the divergent mACP that lacks a Ppant and disfavors association with aACP that retains a Ppant. These features are discussed in the Results section.

Nsf1-mACP-Isd11 interaction and Rieske stability: A native gel and western might help in providing support for complex assembly between mACP-Nsf1-Isd11, and its dissociation in the mutant. This could also be used for further support of the mutagenesis work (Phe111Ala) to study the interaction directly in the parasite.

We agree with the reviewer that further studies of the mACP-Isd11-Nfs1 complex under native conditions will provide deeper insights into its formation and stability. Such studies are in progress.

Some information/data is missing: a detailed list of MS outcome from IPs with mACP and aACP (peptides sequences, peptide numbers, proteins identified, scores); The experiment demonstrating lack of interactions of parasite Isd1 with *E. coli* ACP, along with relevant controls, is not shown; the data supporting the phylogenetic discussion in lines 503-507 is not fully shown.

We provide the key MS data (replicates and spectral counts) relevant for establishing interaction of Nfs1 with mACP over aACP. A full analysis of the IP/MS data for mACP and aACP is beyond the scope of the present manuscript and will be the subject of two separate manuscripts (currently in preparation) that broadly explore the interactomes of mACP and aACP. We plan to publish the full data sets with these manuscripts.

Figure 2E is the key data that establishes lack of interaction of parasite Isd11 with *E. coli* ACP. This figure is a Coomassie-stained SDS-PAGE gel of Ni-NTA pulldown of Isd11 from *E. coli* when co-expressed with parasite mACP. Bacterial ACP has a molecular weight of 9 kDa that is distinct from Isd11 and mACP. The lack of a band for *E. coli* ACP at 9 kDA argues against its association of Isd11. Lack of association of parasite Isd11 with bacterial ACP is consistent with sequence features that suggest that the R6I sequence change in *Plasmodium* Isd11 disfavors association with canonical ACP homologs like *E. coli* ACP that retain a Ppant group (discussed in the Results section). We also subjected the Ni-NTA eluates of Isd11 pulldown to tryptic digest and tandem mass spectrometry. As reported, we did not detect any peptides for bacterial ACP. We are unsure what additional controls the reviewer has in mind. We have further studies in progress to more deeply understand the molecular determinants of ACP-binding specificity by parasite Isd11.

We have added the additional details below to the methods and a supplementary figure to describe our BLAST analysis that only identified divergent mACP homologs in other *Apicomplexa* organisms and in *Vitrella*:

“To identify divergent ACP homologs in other organisms that also lacked the conserved Ser, the amino acid sequence for *P. falciparum* mACP starting with Leu51 was submitted as query to NCBI BLASTP (e-value threshold of 0.05). […] An alignment of these homologs with *P. falciparum* mACP is shown in Figure 6—figure supplement 1.”

A suggestion for enhancing clarity: Some details are hard to find in the manuscript, and might be missing, or need mentioning in context (such as in the relevant point at the result section and within figure legends). For example: what was the cut-off for BLAST search? What do the author mean by "all proteins" in lines 242 and 310? What individual complexes are separated in line 245? What cellular compartment is EF1a from? How many independent repeats are represented by the different Western gels? Is the processing found for mACP in line with any mitochondrial processing predictions?

A) For the BLAST search, only protein hits with e values < 0.01 were retained. We have added this information to table legends describing results of BLAST analyses. We have also moved all methods from the appendix to the main manuscript methods to facilitate access to details such as this one.

B) “All proteins” in these locations refers to the over-expressed recombinant proteins. We have modified the text in both locations to clarify that we refer to the recombinant proteins.

C) Isd11-mACP was the only complex detected in the final purification by size-exclusion chromatography. We have modified the text to clarify.

D) Ef1α is cytosolic (see cited ref. 70). We have modified the text to clearly indicate that this protein is cytosolic.

E) Western blots were repeated 2-3 times. We have modified figure legends and methods to indicate that western blots are representative of 2-3 biological replicates. Biological replicates are included in source data.

F) As discussed in the text, processing of mitochondrial ACP in *Plasmodium* is consistent with known mACP processing in yeast (ref. 28 and 29). We are not aware of any validated software for predicting processing of mitochondrial proteins in parasites.

Reviewer #3 (Recommendations for the authors):Figure 1B (and Figure 1 supplement 3) depict DAPI staining but only Hoechst staining is described in the Materials and methods (line 607).

Hoechst was used for live-parasite imaging while DAPI was used for fixed parasites. We revised the Methods section to clarify this distinction.

Lines 48, 52 and 56 – I'm not a fan of Ppant as an abbreviation for 4-phosphopantetheine. Ppant is not in wide usage (nothing on Google), so why make up your own abbreviations?

We are aware of multiple abbreviations commonly used in the literature for the 4-phosphopantetheine group. Ppant is widely adopted in the bacterial ACP and FASII literatures (e.g., Pubmed 24362570). We prefer this longer, more descriptive abbreviation compared to the less-descriptive 4-PP. In our view, the most critical consideration with abbreviations is to clearly define them, which we do at the beginning of our manuscript.

Line 60 (and lines 510, 512 and 547) – Some general confusion about high level taxon naming conventions. Apicomplexan is an adjective and not a taxon name, so it would only use a capital 'A' when used at the start of a sentence. It also shouldn't be in italics, which are reserved for species names and not used for higher order taxa. You could say Apicomplexa, in which case you would be referring to the Phylum Apicomplexa (no italics). You can also say apicomplexans (no italics), which is an informal category mostly used as equivalent to Phylum Apicomplexa.

We thank the reviewer for this perspective. We have followed the reviewer’s suggestion and modified our use of apicomplexan as an adjective. With respect to italicizing *Apicomplexa* or not, we are not aware of universal agreement or standard on this point. Indeed, compelling arguments have been made to italicize all Latin names to facilitate recognition (e.g., Pubmed 33292779). If *eLife* has an editorial preference in this regard, we would be happy to follow journal standards/norms.

Line 85 – Some readers might take the wrong meaning from this sentence that the ancestral mitochondrial FASII enzymes are now targeted to the apicoplast because of the word 'instead'.

We thank the reviewer for this suggestion. We have revised this sentence to state “In contrast to human and yeast cells, FASII enzymes in *P. falciparum* have been lost by the mitochondrion and are retained instead by the apicoplast organelle.”

Line 90 – a new abbreviation, different to the one in the abstract, for 4-phosphopantetheine is created (4-Ppant). My recommendation is to lose both abbreviations and spell it out.

We feel that abbreviations have value when clearly defined and used in moderation. We have revised the text to exclusively use Ppant.

Lines 134 and 136 – It is molecular mass not weight.

We have revised this description.

Line 141 – molecular mass not MW.

We have revised this description.

Line 146 – This reads as though you somehow induced the bacteria to commence the mACP-HA2 protein with a leucine residue rather than a methionine. Reword. Also, here is a good place to introduce the nomenclature (△2-50) used in Figure 1.

We have re-worded this sentence to read “We therefore cloned and bacterially expressed a truncated (∆2-50) mACP-HA_2_ beginning with Leu-51 (after a start Met).”

Line 165 Figure 1. The DOZI cartoon is perhaps superfluous but if retained, I recommend adding the HA2 tag to the diagram for completeness.

We have added the tag to the diagram.

Line 182 – My version of the file doesn't seem to contain the uncropped blot. Also, no need to abbreviate western blotting to WB.

The uncropped blots are included as Source Data. We feel that abbreviations have value if clearly defined and used judiciously.

Line 198 – 'three-helical' not 3-helical. Spell out numbers less than 10 unless used in conjunction with a unit of measure.

Corrected.

Line 211 – five and 11 amino acids (see above).

Corrected.

Line 302 – 'stabilizes' not stabilize.

The current use of “stabilize” is correct, as the subject is “molecular features”.

Line 338 (and numerous other places) – no need to abbreviate western blotting to WB.

See our response above.

Line 440 – MitoTracker in the text but Mitotracker in Figure 5C and Figure 5 supplement 1.

We have corrected the spelling in Figure 5 and elsewhere.

Line 462 – italicise 'b' in cytochrome b.

Corrected.

Line 464 – add a comma after 'parasites' at the end of the line.

Corrected.

Line 465 – I would qualify this sentence to refer only to the blood stage as follows. 'and few essential functions in the blood stage beyond DHOD and ETC complexes…..;

We have modified this sentence to clarify as the reviewer suggests.

Line 503 – insert comma after FASII and another after Ser on line 504

Corrected.

Line 584 – 'samples'.

Corrected.

Line 587 – 'Triton'.

Corrected.

Line 847 Figure 1 supplement 3 – Is the erythrocyte in the first row infected with two parasites? It seems to have two nuclei and two mitochondria. I consider it best to avoid depicting red cells invaded by multiple parasites as it confuses the uninitiated.

We are uncertain if this image depicts a single parasite in the process of DNA replication/nuclear division or two distinct parasites. The single hemozoin mass is consistent with a single parasite, and the two spatially-distinct mitochondrial signals could be a single tubular mitochondrion passing twice through a single focal plane. The main point of this figure is that signals for mACP and Hsp60 colocalize, as expected for mACP targeting to the mitochondrion.

Line 651 – The References need lots of fixes. i.e. references 1, 17, 21, 31, 34,35, 36,45,49, 50, 60, and 67 all have extraneous capital letters. References 2, 3, 4, 6, 8, 9, 10, 12, 13, 15, 16, 18, 36, 42, 43, 45, 47, 53, 59, 62, and 63 require the Latin genus and species names to be italicised. References 2, 18, 22, and 63 are missing either page numbers or article identifiers.

We have corrected the references. We are happy to further modify if necessary to align with *eLife* requirements.

Line 834 – Figure 1 supplement 1. This is a nice table (and not a figure by the way), but is it relevant? Various alignments such as Figure 1A and Figure 1 supplement figure 2 provide reasonable confirmation that mACP and aACP are two different acyl carrier proteins, and the assignations of the Fab proteins targeted to the apicoplast are well covered in earlier literature.

We feel that this table has value for general readers of *eLife* who may be unaware of the prior literature here. The table is included as a supplement to Figure 1.

The paper uses three letter amino acid abbreviations in the text but single letter amino acid abbreviations in the figures. Consider a consistent system across both.

We use single amino acid abbreviations in figures to minimize use of space. In the text, we use three letter abbreviations when referring to a single amino acid (e.g. Ser) but use single letter abbreviations to refer to series of amino acids (e.g., LYR motif). In our view, this usage optimizes clarity.